# Real-time diagnosis of the hygroscopic growth micro-dynamics of nanoparticles with Fourier transform infrared spectroscopy

Xiuli Wei[1,2], Haosheng Dai[1,2], Huaqiao Gui[1,3*], Jiaoshi Zhang[1], Yin Cheng[1,2], Jie Wang[1], Yixin Yang[1], Youwen Sun[1*], and Jianguo Liu[1,2,3]

1 Key Laboratory of Environmental Optics and Technology, Anhui Institute of Optics and Fine Mechanics, Hefei Institutes of Physical Science, Chinese Academy of Sciences, Hefei 230031, China
2 University of Science and Technology of China, Hefei 230031, China
3 CAS Center for Excellence in Regional Atmospheric Environment, Institute of Urban Environment, Chinese Academy of Sciences, Xiamen 361021, China

*Correspondence to:* hqgui@aiofm.ac.cn (Huaqiao Gui) and ywsun@aiofm.ac.cn(Youwen Sun)

## Abstract

Nanoparticles can absorb water to grow up and this process will affect the light scattering behavior, cloud condensation nuclei properties, lifetime, and chemical reactivity of these particles. Current techniques for calculation of aerosol liquid water content (ALWC) usually restrict the size of particle to be a certain range which may result in a large uncertainty when the particle size beyond the specified range. Furthermore, current techniques are difficult to identify the intermolecular interactions of phase transition micro-dynamics during particles' hygroscopic growth process because their limited temporal resolutions are unable to capture the complex intermediate states. In this study, the hygroscopic growth properties of nanoparticles with electrical mobility diameter ($D_{em}$) of ~100 nm and their phase transition micro-dynamics at molecular level are characterized in real time by using the Fourier transform infrared (FTIR) spectroscopic technique. We develop a novel real-time method for ALWC calculation by reconstructing the absorption spectra of liquid water and realize real-time measurements of water content and dry nanoparticle mass for characterizing the hygroscopic growth factors (GFs). The calculated GFs are generally in good agreement with the extended aerosol inorganics model (E-AIM) predictions. We also explore the phenomenon that the deliquescence points of the ammonium sulfate/sodium nitrate (AS/SN) mixed nanoparticles and the AS/oxalic acid (AS/OA) mixed nanoparticles are lower than that of the pure AS. We further normalize the FTIR spectra of nanoparticles into 2D-IR spectra and identify in real time the hydration

interactions and the dynamic hygroscopic growth process of the functional groups for AS, AS/SN, AS/OA nanoparticles. The results show that both SN and OA compounds can lower the deliquescence point of AS but they affect AS differently. The SN can but OA cannot change the hydrolysis reaction mechanism of AS during the hygroscopic growth process. Compared with previous studies, we captured more complex process and the intermediate state of the hygroscopic growth of nanoparticles. This study can not only provide important information with respect to the difference in phase transition point under different conditions, but also improve current understanding of the chemical interaction mechanism between nanoparticles (particularly for organic particles) and the surrounding medium, which is of great significance for investigation of haze formation in the atmosphere.

**Keywords:** Nanoparticles; phase transition micro-dynamics; Fourier transform infrared spectroscopy; hydration interactions; functional groups

## 1. Introduction

Nanoparticles have long atmospheric lifetimes of weeks to months (Lee and Allen, 2012). As the increase in relative humidity (RH), the sizes of nanoparticles will grow up due to the absorption of water, which may have complex phases and mixing states (Riemer et al., 2019) that influence the light scattering behavior, cloud condensation nuclei properties, lifetime, and chemical reactivity of the nanoparticles (Vogel et al. 2016; Abbott and Cronin, 2021). An improved knowledge of these complex phases and mixing states is crucial for investigating the gas–particle interactions in the atmosphere. Since the nanoparticle size vs. water uptake relationship is influenced by mixing characteristics of various inorganic and organic compounds (Nguyen et al., 2016;Steinfeld and Pandis, 2016), characterization of the water-aerosol interactions is also critical for identifying the fate and transport of trace species in the Earth's system and their effects on air quality, radiative forcing, and regional hydrological cycling (Carlton et al., 2020; Fan et al., 2018).

Ammonium sulfate (($NH_4$)$_2$$SO_4$, AS) is an important atmospheric constituent and a major source of atmospheric nanoparticles originated from anthropogenic activities(Ruehl et al., 2016; Kirkby et al., 2011; Xu et al., 2020). Various techniques

such as the hygroscopic tandem differential mobility analyzer (H-TDMA), the electrodynamic balance (EDB), and the environmental scanning electron microscope (ESEM) have been used to investigate the hygroscopicity of AS (Tang and Munkelwitz, 1977; Tang and Munkelwitz, 1994; Gysel et al.,, 2002; Matsumura and Hayashi 2007). These methods can characterize the deliquescence or phase transition process of particles down to the nanoscale. However, they usually restrict the size of particle to be a certain range in calculation of aerosol liquid water content (ALWC) which may result in a large uncertainty when the particle size beyond the specified range. Furthermore, current techniques are difficult to identify the intermolecular chemical interactions of phase transition micro-dynamics during nanoparticle's hygroscopic growth process because their limited temporal resolutions are unable to capture the complex intermediate states.

Recent studies concluded that the phase transition process of nanoparticles may include multiple intermediate states and are more complex than those disclosed in previous studies. These intermediate states differ from one to the other and last less than 10 ms (Esat et al., 2018). A label-free photonic microscope which uses Bloch surface waves as the illumination source for imaging and sensing is capable to provide real-time measurements of the hygroscopic growth process of a single particle with a diameter of less than 100 nm (Kuai et al., 2020). It can provide valuable insights into the deliquescence and phase transition mechanisms of nanoparticles but cannot identify the chemical composition information of the nanoparticles deliquescence or growth or phase transition process. It is necessary to develop a method to characterize the intermolecular interaction mechanisms of the hygroscopic growth of nanoparticles, which is crucial to understand the physicochemical properties of atmospheric aerosol and the nanoparticle-water interactions during aerosol's hygroscopic growth process. These information are of great significance for improving current knowledge for haze formation.

In this study, the hygroscopic growth properties of pure AS, the $(NH_4)_2SO_4$/NaNO$_3$ (AS/sodium nitrate (SN)) mixed nanoparticles, and the $(NH_4)_2SO_4$/oxalic acid (AS/OA) mixed nanoparticles as well as their phase transition interactions at the molecular level

are characterized in real time by using the Fourier transform infrared (FTIR) spectroscopic technique. We first use a FTIR spectrometer to measure and an extended aerosol inorganics model (E-AIM) to predict the hygroscopic growth factors (GFs) of AS, AS/SN and AS/OA nanoparticles. We further normalize the FTIR spectra of nanoparticles into 2D-IR spectra to analyze the intermolecular interactions during the hygroscopic growth processes of these nanoparticles. This study can not only provide important information with respect to the difference in phase transition point under different conditions, but also improve current understanding of the chemical interaction mechanism between nanoparticles (particularly for organic particles) and the surrounding medium, which is of great significance for investigation of haze formation in the atmosphere.

## 2. Material and method

### 2.1 Experiment description

The experimental setup includes a nanoparticle generation system, a humidification system, and a FTIR analysis system. Particles are aerosolized by an atomizer (model 255, MetOne), dried by a diffusion dryer (model 3062, TSI), sorted into a specific electrical mobility diameter ($D_{em}$) by a differential mobility analyzer (DMA; model 3082, TSI), and finally deposited onto a 3 cm × 3 cm zinc selenide (ZnSe) substrate inside a cylinder sample cell with a radius of 3 cm and a length of 4 cm through a cone-shaped hole (Figure 1). The sheath-to-sample flow ratio of the DMA is set to be 10:1 (the sheath flow is 10 L/min and the sample flow is 1 L/min), which can produce an effective mobility for the measured aerosols with sizes ranging from 14.9 to 673.2 nm. We only selected nanoparticles with $D_{em}$ of ~100 nm for deposition. After a deposition time of ~12 h, the substrate is sealed inside the sample cell to obtain a stable RH condition for subsequent analysis. There are about 100 thousand nanoparticles deposited onto the substrate. For the 100 nm nanoparticle, its hydration characteristic mainly depends on its chemical composition and the kelvin effect is negligible(Cruz and Pandis, 2000; Lee et al., 1998). The enrichment for the nanoparticles is to improve the signal of FTIR measurement because the hygroscopic signal of a single nanoparticle is too weak to be measured by the FTIR method. Since the chemical composition is not changed during the deposition, this deposition process can obtain the same results as those for a single particle.

The humidification system can provide a specific RH for the sample cell (Kuai et

al., 2020). A RH sensor (HC2-S, Rotronic Incorporation, Switzerland) with an accuracy of ± 0.8% for 0 -100% RH range is mounted at the inlet of the sample cell. The RH downstream of the DMA varies over 16.52 to 18.74% which are well below the efflorescence relative humidity (ERH) of AS (about ∼32% RH) (Figure S1). As a result, the initial states of all nanoparticles are in dry conditions.

The FTIR spectrometer (Tensor 27, Bruker Optics, Germany) starts to take absorption spectra of the samples approximately 5 min after the injection of each designated RH. This time interval is used to stabilize the atmospheric condition, especially the RH, inside the sample cell. The FTIR spectrometer is equipped with a KBr beam splitter and a liquid nitrogen-cooled mercury cadmium telluride (MCT) detector for measuring the absorption spectra of the samples. A He-Ne laser metrology keeps the FTIR instrument in a good optical alignment. The FTIR spectrometer saves middle infrared (MIR) spectra from 800 to 4000 $cm^{-1}$ with a spectral resolution of 4 $cm^{-1}$ and a repeat times of 64. An air conditioner is run uninterruptedly to keep the laboratory under a constant temperature of ~25 °C.

## 2.2 Sample description

In this study, all chemical reagents are produced by the Aladdin Reagent Inc. (reagent grade, 99.8% purity), and the water is obtained from an ultrapure water system (Direct-Q3, Millipore). Table 1 summarizes all chemical compounds and their concentrations used in the experiment. The density, solubility, and molecular mass of all chemical compounds are prescribed from the Handbook of Chemistry and Physics (Lide, D. R., 2007). All single chemical compounds are dissolved individually in ultrapure water with a concentration of 4.0 g/L. All mixed solutions (including AS/SN and AS/OA) are generated by mixing the two corresponding single compounds with a mass ratio of 1:1. As a result, both AS/SN and AS/OA nanoparticles are internally mixed nanoparticles.

AS is selected as a representative of inorganic salt and OA is an important water-soluble organic compound contained in atmospheric aerosols. We select AS as a representative of inorganic salt because it is a significant constituent of the submicron-scale aerosol in the atmosphere. In addition to an important water-soluble organic compound contained in atmospheric aerosols (Wang et al., 2019), OA is also the dominant dicarboxylic acid in both urban and remote atmospheric aerosols (Richards et al., 2020).

## 2.3 Methodology

### 2.3.1 Quantifying ALWC and the mass of nanoparticles

We first correct the baseline of the measured absorption spectra with the Opus 7.0 software provided by Bruker, Germany. We then iteratively recalculate the spectra with the absorption coefficients of liquid water provided by Downing and Williams (Downing and Williams, 1975) through the non-linear least squares method till the residual between the measured spectra and the calculated spectra are minimized. We stop the iteration and derive the liquid water content once the root mean square error (RMSE) of the residual is below 0.3%. After the deliquescence of the nanoparticles and the position of the absorption peak (referring to the wavenumber that shows the strongest absorption) of $SO_4^{2-}$ is relative stable, we use a similar non-linear least square method to derive the mass of $SO_4^{2-}$ ($M_{sulfate}$) (Wei et al., 2019). Finally, the mass of AS ($M_{AS}$), AS/SN ($M_{AS/SN}$), and AS/OA ($M_{AS/OA}$) can be derived with the $M_{sulfate}$ via the following equations.

$$M_{AS} = \frac{M_{sulfate}}{96} * 132 \tag{1}$$

$$M_{AS/SN} = \frac{M_{sulfate}}{96} * 132 * 2 \tag{2}$$

$$M_{AS/OA} = (\frac{M_{HSO_4^-}}{97} + \frac{M_{sulfate}}{96}) * 132 * 2 \tag{3}$$

### 2.3.2 GFs calculation

The GF indicating the water uptake ability of aerosol particles is defined as GF = $D_{wet}/D_0$, where $D_{wet}$ (cm) is the mean volume equivalent diameter ($D_{ve}$) of the particles at the designated RH and $D_0$ (cm) is the mean initial $D_{ve}$ of the dry particles at the temperature of the sample cell. In present work, the RH varies from 50% to 95% and we assume that the temperature of the sample cell equals to the room temperature of ~ 25°C . The GF used for investigation of hygroscopic growth properties of nanoparticles can be calculated via equations (4) to (7),

$$V_{water} = \frac{M_{water}}{\rho_{water}} \tag{4}$$

$$V_0 = \sum_i (\frac{M_i}{\rho_i}) = \sum_i (V_i) \tag{5}$$

$$V_{wet} = V_0 + V_{water} \tag{6}$$

$$GF \; = \; \frac{D_{wet}}{D_0} = (\frac{V_{wet}}{V_0})^{1/3} \tag{7}$$

Where $V_0$ (cm$^3$) is the initial volume of the dry nanoparticle at approximately 25°C and $V_{water}$ (cm$^3$) is the volume of water contained in the nanoparticle at the designated RH. Yan et al. (2020) have compared the $D_{ve}$ and the $D_{em}$ of AS sorted by the identical DMA of this study. A good agreement between $D_{em}$ and $D_{ve}$ for ~100 nm AS was observed by Yan et al. (2020). As a result, in present work, we use $D_{em}$ the same as the $D_{ve}$. $M_{water}$ (g) is the calculated water content at the designated RH; $M_i$ (g) is the calculated mass of the $i$th pure compound; $\rho_{water}$ (g/cm$^3$) (approximately 1 g/cm$^3$) and $\rho_i$ (g/cm$^3$) are the densities of water and the $i$th pure compound, respectively, and $i$ is the number of pure compound.

We use the Extended Aerosol Inorganic Model (E-AIM) proposed by Wexler and Clegg to predict the GFs of nanoparticles (http://www.aim.env.uea.ac.uk/aim/aim.php). The E-AIM takes the solution thermodynamics into consideration, including the water activity, the phase state, and the equilibrium distribution in the particles. The E-AIM calculate the water activity of the organic water mixtures based on the contributions of the functional groups.

### 2.3.3 The 2D-IR analysis method

Although the absorption spectra recorded by the FTIR spectrometer can be used to characterize the liquid water content and the mass of functional groups contained in the nanoparticles during the hygroscopic growth process, the absorption peaks of the nanoparticles (especially for organic compounds) are difficult to separate since they are overlapped with each other. In contrast, the 2D-IR analysis technique can resolve the overlapped absorption peaks (McKelvy et al., 1998; Du et al., 2021) and, more importantly, can provide detailed information about the dynamic hygroscopic growth process of the functional groups (Noda and Ozaki, 2014).

After baseline correction, we normalize all infrared spectra into 2D-IR spectra (denoting as D) with the 2D Shige software developed by Kwansei-Gakuin University, Japan (Noda and Ozaki, 2014). The 2D-IR spectra represent the perturbation-induced variations of a series of spectral intensity observed during the interval of external variable RH. As expressed in equations (8) and (9), the 2D-IR spectra can be used to calculate the synchronous and asynchronous correlation coefficients of the spectral intensities at different wavenumbers. The wavenumber regions ranging from 800–

1400cm$^{-1}$ and from 2800–3800cm$^{-1}$ which almost cover the absorption features of all identifiable functional groups of interest are selected for analysis.

$$\Phi(\nu_1, \nu_2) = D^T * D \qquad (8)$$

$$\Psi(\nu_1, \nu_2) = D^T * HD \qquad (9)$$

where $D^T$ denotes as the transposed D, $\Phi(\nu_1, \nu_2)$ and $\Psi(\nu_1, \nu_2)$ represent the synchronous and asynchronous correlation coefficients of the spectral intensities at the wavenumber $\nu_1$ and $\nu_2$, respectively. $\Psi(\nu_1, \nu_2)$ is obtained by orthogonalizing D with the Hilbert transform matrix H and calculating the rows cross-product between D and the orthogonal matrix HD. The synchronous map displays correlations between all spectral intensities changing in phase in the experiment and shows whether they increase or decrease relative to each other. The asynchronous correlation map, in contrast, relates spectral intensities that change at different rates and contain information about the sequence of the occurring events. In this study, we use the synchronous correlation maps to diagnose if the spectral intensities at different wavenumbers vary simultaneously, and use the asynchronous correlation maps to identify the occurrence sequential order of the hydration interactions.

In present work, the red and blue areas in the 2D-IR spectra indicate positive and negative correlations of the spectral intensities at $\nu_1$ and $\nu_2$, respectively. In the synchronous correlation maps, the positive and negative correlations indicate simultaneous and opposite changes of the spectral intensities observed at the wavenumber pair ($\nu_1$, $\nu_2$), respectively. In the asynchronous correlation maps, the positive correlation indicates the spectral intensity change at $\nu_1$ occurs predominantly before that at $\nu_2$, while the negative correlation indicates the spectral intensity change at $\nu_2$ occurs predominantly before that at $\nu_1$.

## 3. Results and discussion

### 3.1 Spectral characteristics of nanoparticles during hygroscopic growth process.

Figure 2 shows the FTIR spectral absorption characteristics of the AS nanoparticles under humidity conditions from 50% to 90%. Figure 3 shows the predicted $M_{water}/M_0$ and the measured hygroscopic properties of 100 nm AS nanoparticles as a function of RH during the hygroscopic growth process. The strong absorption peaks observed at 3250 cm$^{-1}$ and 1112 cm$^{-1}$ at the initial RH of 50% are the stretching vibration peak of

OH and the symmetrical stretching vibration peak of $SO_4^{2-}$, respectively(Wang et al., 2017; Nájera and Horn, 2009; Gopalakrishnan et al., 2005). The areas of OH and $SO_4^{2-}$ absorption peaks reflect the liquid water content and the concentration of $SO_4^{2-}$ contained in the AS nanoparticles. With the increase in RH between 50% and 79%, the $SO_4^{2-}$ absorption peak (1112 cm$^{-1}$) starts to redshift slowly (Figure 3), which indicates that liquid water molecules have been attached to the surface of the solid AS nanoparticles, and the $SO_4^{2-}$ is then bonded with these liquid water molecules to form a hydrogen bond during this hygroscopic growth process (Yeşilbaş and Boily, 2016). In the meantime, the position and the area of the OH absorption peak don't change significantly, indicating no hygroscopic growth of the AS nanoparticles occurs for the RH between 50% and 79% (Wang et al., 2019; Tang et al., 2016; Nájera and Horn 2009; Martin 2000). It is worth noting that the $SO_4^{2-}$ has two different absorption peaks in solid and aqueous AS nanoparticles and Figure 3 only presents the area of $SO_4^{2-}$ absorption peak in aqueous AS nanoparticles. As a result, it is zero for the RH between 50% and 79% since no hygroscopic growth occurs in this RH range.

When the RH reaches 79%, the position of the $SO_4^{2-}$ absorption peak has shifted from 1102 to 1097 cm$^{-1}$ and its area increases abruptly from 0 to 0.38. In the meantime, the position of the OH absorption peak is still the same as that in initial humidity condition (50%) but its area increases abruptly from 0.18 to 0.67 (Figure 3). The area of the OH absorption peak can be used to ascertain the phase transition of nanoparticles since it is sensitive to surrounding chemical environment (Braban et al., 2003). The abrupt increase in the area of the OH absorption peak indicated the phase transition of AS, i.e., the AS nanoparticles have absorbed water rapidly and transformed from the crystalline phase to the aqueous phase. According to the E-AIM predictions and the results from previous studies (Estillore et al. , 2016; Cruz and Pandis, 2000; Tang 1982), this process is called the deliquescence, and the RH at this stage is referred to as the deliquescence RH (DRH). When deliquescence occurs, $NH_4^+$ molecules hydrated with $SO_4^{2-}$ are replaced with $H_2O$ molecules, which leads to the redshift of the $SO_4^{2-}$ absorption peak (Dong et al., 2007). Tang et al. (1982), Cruz and Pandis (2000), and Estillore et al. (2016) have used the photonic microscope to observe the hygroscopic growth properties of large-size AS particles. Our method and the particle size are different from previous studies, but we obtained a consistent DRH (about 79% $\pm$ 0.8%) with those in Tang et al. (1982), Cruz and Pandis (2000), and Estillore et al. (2016),

where the DHR for AS was found to be 79% $\pm$ 1% or 80% $\pm$ 0.4%. This is because the hydration characteristics of nanoparticles mainly depend on their chemical composition and the kelvin effect is negligible. Lee et al. (1998) concluded that, for particles with $D_{ve}$ of larger than 100nm, their critical hydration characteristics are essentially independent of the particle size and are similar to the condensation of water on the flat surface (Lee et al., 1998).

The AS nanoparticles continue to be humidified after deliquescence, resulting in a further increase in the area of the OH absorption peak due to continuous water uptake. However, the position of the $SO_4^{2-}$ absorption peak keeps constant regardless of RH, indicating that the AS nanoparticles are still in the aqueous phase after deliquescence. As a further increase in RH, the volume of AS nanoparticles increases due to the increase in liquid water content, but the mass of AS nanoparticles keeps constant, resulting in a decrease in $SO_4^{2-}$ concentration. As a result, we observe a decrease in the area of $SO_4^{2-}$ absorption peak starting from ~ 83% RH.

Figure 4 is same as Figure 2 but for the OA nanoparticles. The results for the OA nanoparticle differ from those for AS. The FTIR spectral absorption characteristics of SN, AS/SN, AS/OA nanoparticles under humidity conditions from 50% to 90% are shown in Figure S2, Figure S3, Figure S4, respectively.Throughout its hygroscopic growth process, the OH absorption peak at 3250 cm$^{-1}$ was not detected, indicating that liquid water is not absorbed by the OA. Previous studies conclude that, for the dihydrate crystalline state of OA particle, its deliquescence point is larger than 97% RH (Peng et al., 2001), but for the amorphous state of OA particle, it starts to water uptake above 45%RH (Mikhailov et al., 2009). From this point of veiw, the OA particles in this study could be in the state of dihydrate crystalline state and our findings are constituent with those of Jing et al., (2016) and Ma et al. (2019 ).

Figure 5 compares the predicted $M_{water}/M_0$ ($M_{water}$ is the mass of liquid water in the nanoparticles; $M_0$ is the initial mass of nanoparticles) ratio by the E-AIM (UNIFAC model) (http://www.aim.env.uea.ac.uk/aim/aim.php) and the measured hygroscopic properties from the FTIR spectra with the method described in section 2.3 for the AS, AS/OA, and AS/SN nanoparticles during the hygroscopic growth process from 50% to 90% (RH). The results show that the predicted and measured $M_{water}/M_0$ results for both pure and mixed nanoparticles are generally in good agreement during the whole hygroscopic growth process, indicating consistent water uptake between the predictions

and the measurements with the increase in RH.

**3.2 Hygroscopic growth properties of pure and mixed nanoparticles**

With the results derived from the FTIR measurements, we calculated the GFs for both pure and mixed nanoparticles via equation (7) and investigated their variabilities with respect to the changes in RH. Figure 6 compares the measured and predicted GFs for both pure and mixed nanoparticles under the humidity conditions from 50% to 90%. The measured and predicted GFs are in good agreement through the whole humidity range. The GFs can be obtained precisely using the H-TDMA technique via a direct measurement of the aerosol diameter. In this study, the GFs for both pure and mixed nanoparticles are calculated with liquid water content and the relative masses of dry nanoparticles obtained from FTIR measurements. Though with different methods, the GFs for both pure and mixed nanoparticles in this study are in good agreement with those from previous studies (Jing et al., 2016; Braban et al,. 2003).

The GF curves can be used to investigate the sensitivity of particle volume to RH. At the RH of 79 $\pm$0.8%, deliquescence occurs, the diameter of AS nanoparticle grows up sharply by up to approximately 1.48 times and transforms from the crystalline to aqueous phase. At the RH of 85 $\pm$0.8%, the GF for AS is 1.65$\pm$0.05 which is a slightly higher than the values deduced with the H-TDMA (about 1.49, Cruz and Pandis, 2000) and Environmental Scanning Electron Microscope (about 1.50, Matsumura and Hayashi 2007) but is close to the value deduced with the E-AIM (about 1.60).

Since both the AS/SN and AS/OA mixed nanoparticles have lower DRHs than that of the AS and absorb liquid water below their DRHs, their GF curves differ from the pure AS particle. Although OA particle does not absorb water, OA and AS in the AS/OA aqueous solution can react with each other via the following pathway (Minambres et al., 2013):

$$(NH_4)_2SO_4 + H_2C_2O_4 \rightarrow NH_4HSO_4 + NH_4HC_2O_4 \tag{10}$$

This reaction can be identified in Figure S5, where the absorption peak at 1245cm$^{-1}$ is the stretching vibration peak of HSO$_4^-$. As a result, a lower DRH for the mixed nanoparticles relative to the pure AS can be attribute to the formation of NH$_4$HSO$_4$ which has a lower DRH (about 40%) than the pure AS (80%) (Tang and Munkelwitz, 1994). All above findings are in good agreement with those in Seinfeld and Pandis. (2016) which measure GF with H-TDMA.

### 3.3 Phase transition dynamics of pure AS nanoparticles

The 2D-IR spectra based on the spectral variations induced by the deliquescence of pure AS between 50–90% RH are shown in Figure 7. The correlation maps for the OA nanoparticles are not shown because they don't absorb water and thus present no deliquescence transition during hygroscopic growth process. In the synchronous correlation maps, one main red/positive (3250, 1097) auto-peak was observed for the

AS nanoparticles in the 50–90% RH range, which indicated that the simultaneous increases in the spectral intensities of OH and $SO_4^{2-}$ absorption peaks. This means that the liquid water and $SO_4^{2-}$ in the aqueous AS nanoparticles increase simultaneously during hygroscopic growth process. Furthermore, two main blue/negative (1112, 1097) and (3250, 1112) auto-peaks were also observed for the AS nanoparticles. The

blue/negative (1112, 1097) auto-peak indicated that the spectral intensity of the OH absorption peak increases over RH, while the spectral intensity of the $SO_4^{2-}$ absorption peak in the solid AS nanoparticles decreases over RH. The blue/negative (1112, 1097) auto-peak indicated that the $SO_4^{2-}$ in solid and aqueous AS nanoparticles can transform into each other, and the decrease of $SO_4^{2-}$ in solid AS nanoparticles results in the

increase of $SO_4^{2-}$ in aqueous AS nanoparticles. This behavior can be explained by the fact that $NH_4^+$ particles hydrated with $SO_4^{2-}$ are replaced by $H_2O$ molecules with the increase in RH.

The asynchronous map indicates the sequential changes in the spectral intensities in response to the hygroscopic activities. As the asynchronous correlation maps shown

in Figure 7, two main red/positive (3250, 1097) and (1112, 1097) and two main blue/negative (3250, 1112) and (1097, 1112) auto-peaks were observed for the AS nanoparticles, which indicated that spectral intensities of the absorption peaks changed in the order from (1112) cm$^{-1}$ > (3250) cm$^{-1}$ > (1097) cm$^{-1}$. The decrease of $SO_4^{2-}$ in the solid AS nanoparticles does not result in a simultaneous increase of $SO_4^{2-}$ in the

aqueous AS nanoparticles during the AS deliquescence transition. The former process occurred predominantly before the later process. It suggests an intriguing possibility of the existence of an intermediate state between the solid and aqueous AS nanoparticles. Meanwhile, the water uptake occurred predominantly before the decrease of $SO_4^{2-}$ in the solid AS nanoparticles during the AS deliquescence transition. We speculate that

the surface-limited process may control the transport of liquid water to the AS nanoparticle, or in other words, the surface limited process determine the hygroscopic

behavior of AS nanoparticles (Leng et al., 2015).

The above 2D-IR spectroscopic results verify that the hygroscopic growth of AS nanoparticle may include the following phase transition micro-dynamics stages at molecular level: the first stage (adsorption) pertains to the attachment of liquid water molecules to the surface of the solid $(NH_4)_2SO_4$ (Yeşilbaş and Boily, 2016), which would cause a decrease of the solid $(NH_4)_2SO_4$; then the $NH_4^+$ particles hydrated with $SO_4^{2-}$ are gradually replaced by the $H_2O$ molecules and finally, the AS nanoparticles become fully liquid droplets. Although the hygroscopic growth characteristics observed in this study are similar to those in previous study (Cruz and Pandis, 2000), the 2D-IR spectroscopic technique captured more complex process and the intermediate state during the hygroscopic growth of AS nanoparticles (Tang, et al., 1977; Tang, et al. 1994).

### 3.4 Phase transition dynamics of mixed nanoparticles

Figure 8 is the same as Figure 7 but for AS/SN mixed nanoparticles. In the synchronous correlation maps, one main red/positive (3250, 1097) and two blue/negative (1320, 1112) and (3250, 1112) auto-peaks were observed in the RH range from 50–90%. The red/positive (3250, 1097) auto-peak indicated the simultaneous increase in the spectral intensities of OH and $SO_4^{2-}$ absorption peaks. The blue/negative (1320, 1112) and (3250, 1112) auto-peaks indicated that the spectral intensities of the $NO_3^-$ and the OH absorption peaks increase over RH, while the spectral intensity of $SO_4^{2-}$ absorption peak in solid AS/SN mixed nanoparticles decreases over RH. This can be explained by the fact that the $NH_4^+$ or $Na^+$ hydrated with $SO_4^{2-}$ and $NO_3^-$ are replaced by the $H_2O$ molecules with the increase in RH. In the asynchronous correlation map, three main red/positive (3250, 1097), (1097, 1112), and (1320, 1097) and two main blue/negative (3250, 1320) and (1112, 1097) auto-peaks were observed, which indicated that the spectral intensities of the absorption peaks change in the order from (1320) cm$^{-1}$ > (3250) cm$^{-1}$ > (1097) cm$^{-1}$ > (1112) cm$^{-1}$. This means that the increase of $NO_3^-$ occurred predominantly before the increase of $SO_4^{2-}$ in the aqueous AS/SN mixed nanoparticles because the SN has a lower DRH (RH=74.3±0.4%) relative to that of AS (Seinfeld and Pandis, 2016; Tang and Munkelwitz, 1993)). Meanwhile, the decrease of $SO_4^{2-}$ in solid AS/SN mixed nanoparticles occurred predominantly after the increase of $SO_4^{2-}$ in the aqueous AS/SN mixed nanoparticles, which suggests an intermediate state between the aqueous $SO_4^{2-}$ and solid $SO_4^{2-}$ states. Furthermore, the

hydrolysis reaction mechanism for the AS in AS/SN mixed nanoparticle differs from that for the pure AS nanoparticle. This is because $NO_3^-$ absorb water at a lower RH than that of pure AS nanoparticle, which enhances the dissolution of the pure AS in the AS/SN mixed nanoparticle. Therefore, the $NH_4^+$ hydrated with the $SO_4^{2-}$ are replaced by the $H_2O$ molecules. This process continues till the AS/SN mixed nanoparticle become fully liquid droplets (Jing et al., 2016).

Figure 9 is the same as Figure 8 but for the AS/OA mixed nanoparticles. In the synchronous correlation map, one main red/positive (3250, 1080) and two main blue/negative (1112, 1080) and (3250, 1112) auto-peaks were observed in the RH range from 50–90%. The wavenumber of 1080 $cm^{-1}$ is the $SO_4^{2-}$ absorption peak in the aqueous AS/OA mixed nanoparticles. The red/positive (3250, 1080) auto-peak indicated the simultaneous increase in the spectral intensities of OH and $SO_4^{2-}$ absorption peaks. The blue/negative (1112, 1080) auto-peak indicated that the $SO_4^{2-}$ in solid and aqueous AS/OA nanoparticles can transform into each other, and the decrease of $SO_4^{2-}$ in solid AS/OA nanoparticles results in the increase of $SO_4^{2-}$ in aqueous AS/OA nanoparticles. The blue/negative (3250, 1112) auto-peak indicated that the spectral intensity of the OH absorption peak increases over RH, while the intensity of the $SO_4^{2-}$ absorption peak in solid AS/OA nanoparticles decreases over RH. In the asynchronous correlation map, one main red/positive (3250, 1080) and one main blue/negative (3250, 1112) auto-peaks were observed, which indicated that the spectral intensities of the absorption peaks change in the order from $(1112)\,cm^{-1} > (3250)\,cm^{-1} >$ $(1080)\,cm^{-1}$. This occurrence sequential order is consistent with that of the pure AS nanoparticle during the hygroscopic growth process. This is because OA does not absorb water, and thus the hydrolysis reaction mechanism of AS in AS/OA mixed nanoparticle is similar to that of the pure AS nanoparticle. With current measurements, we cannot judge if the hygroscopic growth process of AS and AS/OA have the same intermediate states.

## 4. Conclusions

In this work, we demonstrated the usage of FTIR spectroscopic technique to characterize in real time the hygroscopic growth properties of nanoparticles with electrical mobility diameter ($D_{em}$) of ~100 nm and their phase transition micro-dynamics at molecular level. We first realize real-time measurements of water content and dry nanoparticle mass for characterizing the hygroscopic growth factors (GFs). The

calculated GFs are generally in good agreement with the extended aerosol inorganics model (E-AIM) predictions in the 50–95% RH range. We further normalize the FTIR

spectra of nanoparticles into 2D-IR spectra and identify in real time the hydration interactions and the dynamic hygroscopic growth process of the functional groups for AS, AS/SN, AS/OA nanoparticles. The 2D-IR spectroscopic results disclosed that the hygroscopic growth of nanoparticle may include the following phase transition micro-dynamics stages at molecular level:

a) For pure AS nanoparticles: The first stage (adsorption) pertains to the attachment of liquid water molecules to the surface of the solid $(NH_4)_2SO_4$, which would cause a decrease of the solid $(NH_4)_2SO_4$; then the $NH_4^+$ particles hydrated with $SO_4^{2-}$ are gradually replaced by the $H_2O$ molecules and finally, the AS nanoparticles become fully liquid droplets.

b) For the AS/SN: The hydrolysis reaction mechanism for the AS in AS/SN mixed nanoparticle differs from that for the pure AS nanoparticle. The increase of $SO_4^{2-}$ in the aqueous AS/SN mixed nanoparticles occurred predominantly before the decrease of $SO_4^{2-}$ in solid AS/SN mixed nanoparticles. Then, the $NH_4^+$ hydrated with the $SO_4^{2-}$ are replaced by the $H_2O$ molecules. This process

continues till the AS/SN mixed nanoparticle become fully liquid droplets (Jing et al., 2016).

c) For AS/OA mixed nanoparticles: The hydrolysis reaction mechanism of AS in AS/OA mixed nanoparticle is similar to that of the pure AS nanoparticle. With current measurements, we cannot judge if the hygroscopic growth process of

AS and AS/OA have the same intermediate states.

Although the hygroscopic growth characteristics observed in this study are similar to those in previous study, the FTIR spectroscopic technique demonstrated in this study captured more complex process and the intermediate state during the hygroscopic growth of nanoparticles. This study verified that the FTIR spectroscopic technique

provides a new suitable method for real-time diagnosis of the hygroscopic growth micro-dynamics of nanoparticles at molecular level. By means of this new method, we can better understand the physicochemical properties of atmospheric aerosol and the nanoparticle-water interactions during aerosol's hygroscopic growth process. These information are of great significance for improving current knowledge for haze

formation.

## Author contribution

XW designed the experiment and wrote the paper with contributions from all co-authors; HG contribute to science discussions and suggested analyses; HD and JZ prepared for the humidification system; YC, JW, YY and JL contributed to this work by providing constructive comments; YS contributed to this work by providing constructive comments, review, and editing.

## Competing interests

The authors declare that they have no conflict of interest that could have appeared to influence the work reported in this paper.

## Acknowledgments

This work was supported by the National Natural Science Foundation of China (No. 41905028, 91544218), the Natural Science Foundation of Anhui (No. 2108085MD139), the Science and Technological Fund of Anhui Province for Outstanding Youth (No.1808085J19). We are also grateful to the China Scholarship Council for their support.

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

# Table

Table 1. The density, solubility, and molecular mass of all chemical compounds used in this study

| Chemical compound | Molecular mass (g/mol) | Density (g/cm³) | Solubility in $H_2O$ at 25 °C (g/100 cm³) |
|---|---|---|---|
| $(NH_4)_2SO_4$ (AS) | 132.14 | 1.769 | 75.4 |
| $NaNO_3$ (SN) | 84.99 | 2.257 | 88 |
| Oxalic acid (OA) | 90.04 | 1.900 | 9.52 |

# Figures

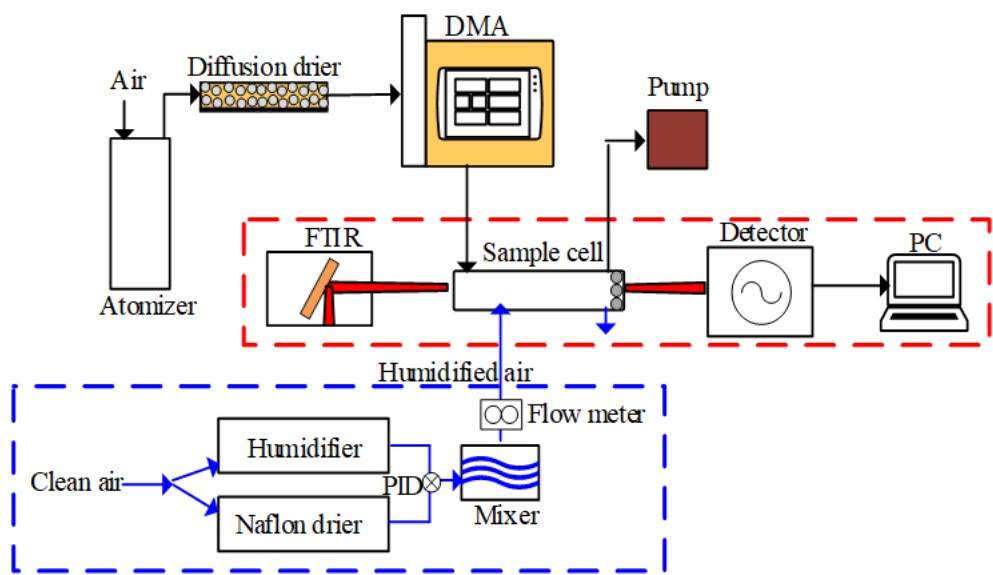

**Figure 1.** Diagrammatic sketches of the experimental system used to measure nanoparticle hygroscopicity. The red dashed box represents the FTIR system and the blue dashed box represents the humidification system (DMA: differential mobility analyzer; PID: proportional integral differential control).

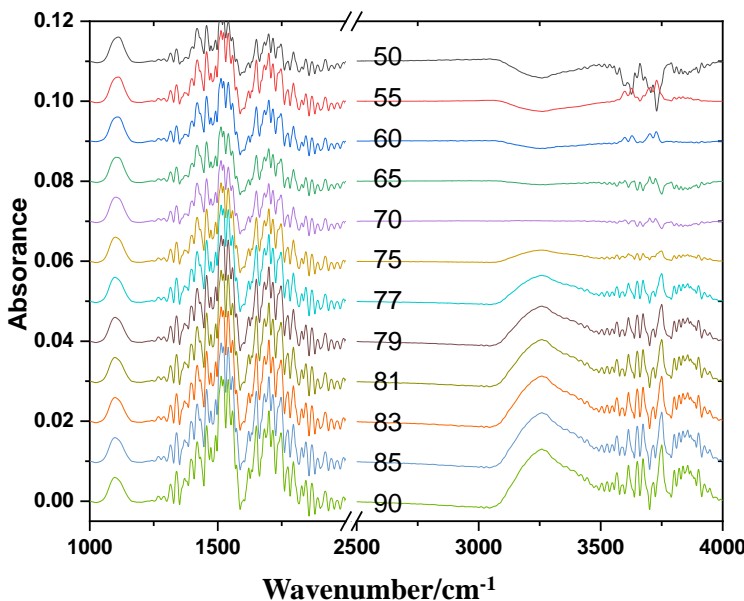

**Figure 2.** FTIR spectral characteristics of the AS nanoparticles under humidity conditions from 50% to 90% (RH).

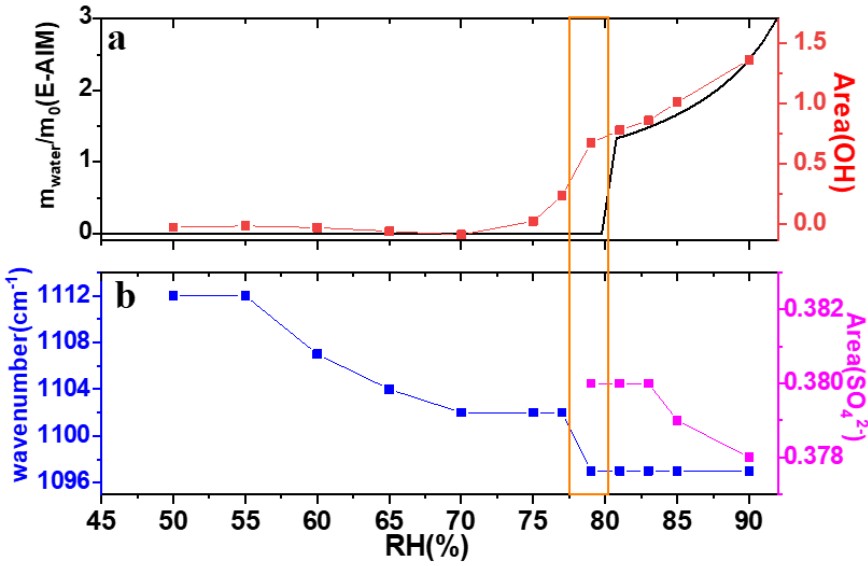

690

**Figure 3.** (a) Predicted $M_{water}/M_0$ and the area of OH stretching peak as a function of RH; (b) The center wavenumber and the area of the symmetrical stretching vibration peak of $SO_4^{2-}$ in the aqueous AS nanoparticles as a function of RH. The black curves show the E-AIM predictions. The orange box indicates the deliquescence relative humidity point.

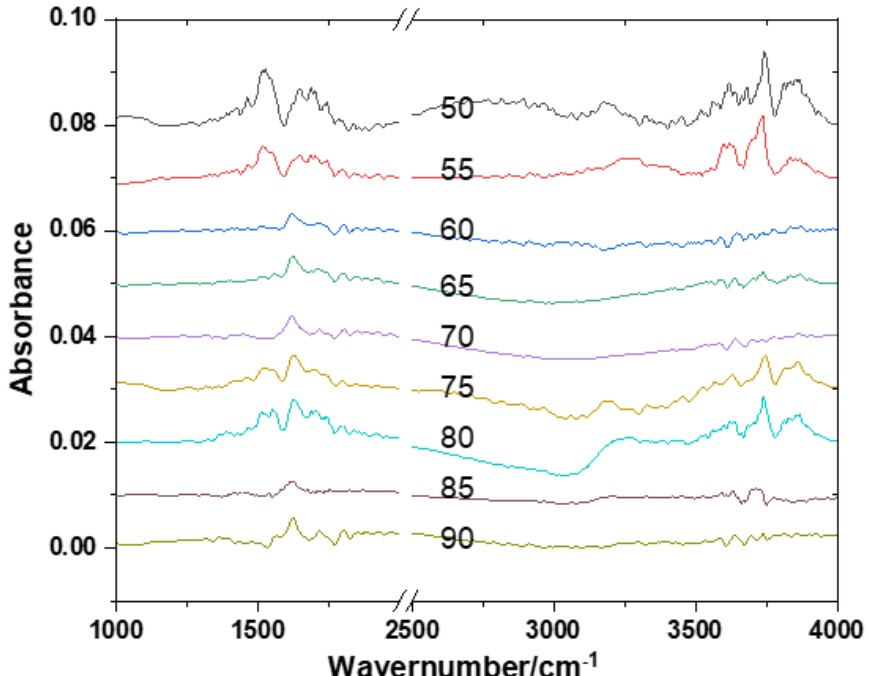

695

**Figure 4.** The same as Figure 2 but for OA nanoparticles.

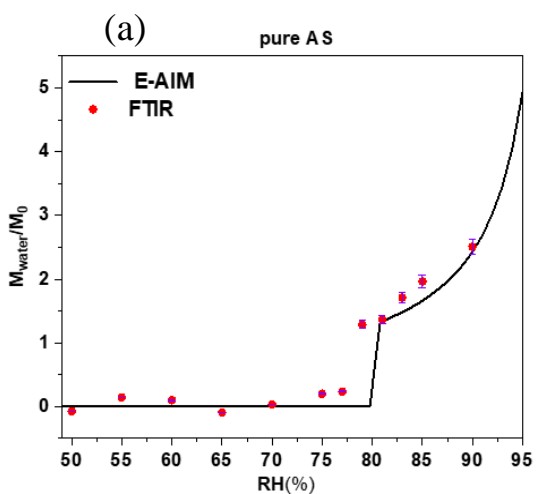

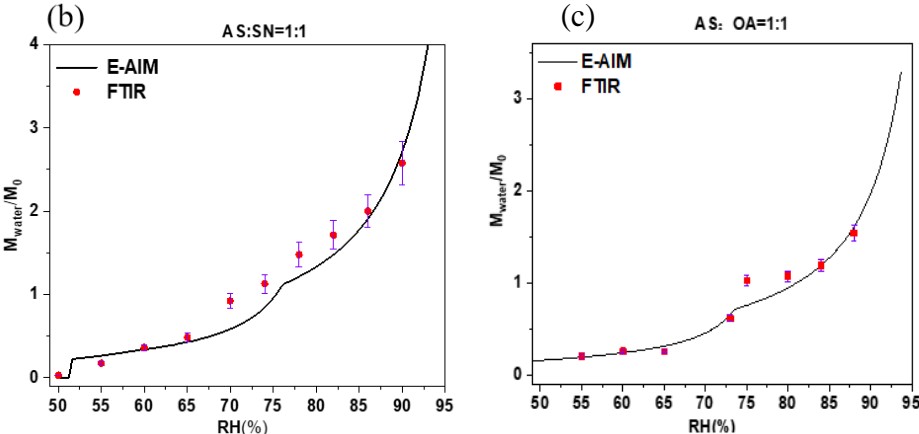

**Figure 5.** Comparison between the measured and predicted $M_{water}/M_0$ for pure AS (a), AS/OA (b), and AS/SN nanoparticles (c) with a dry diameter of 100 nm during the hygroscopic growth process as a function of the RH. The black curves represent the E-AIM predictions and the red square represent the measured results from the FTIR spectra. The error bar is defined as the $1\sigma$ standard deviation of repeated measurements.

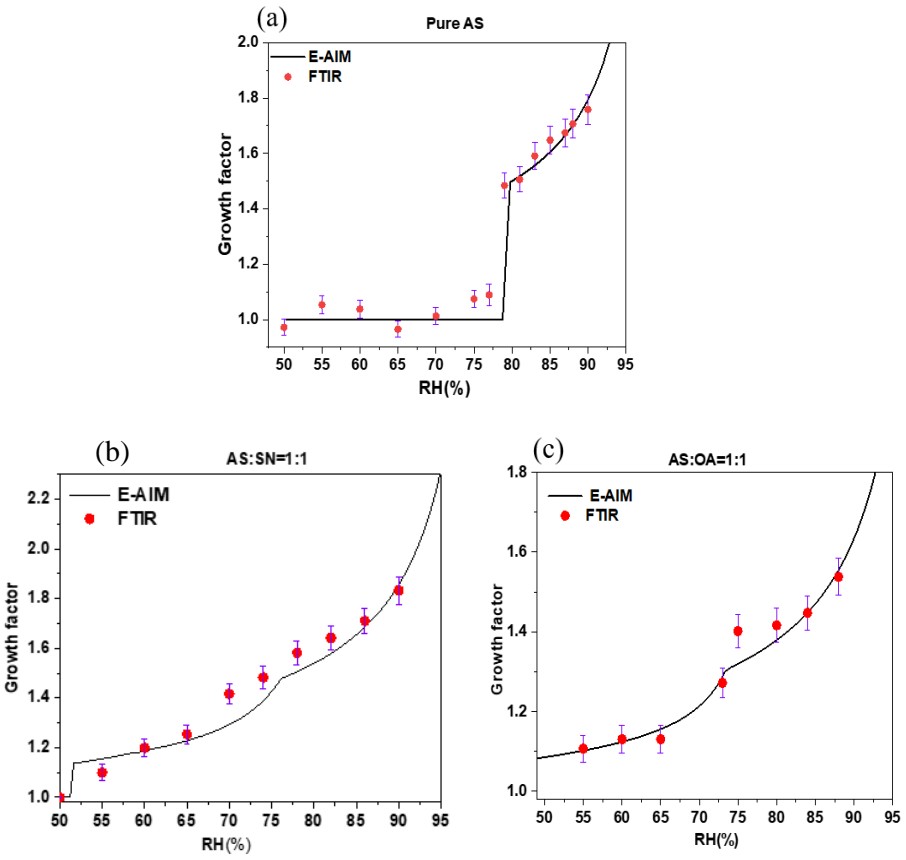

**Figure 6** The same as Figure 5 but for GFs.

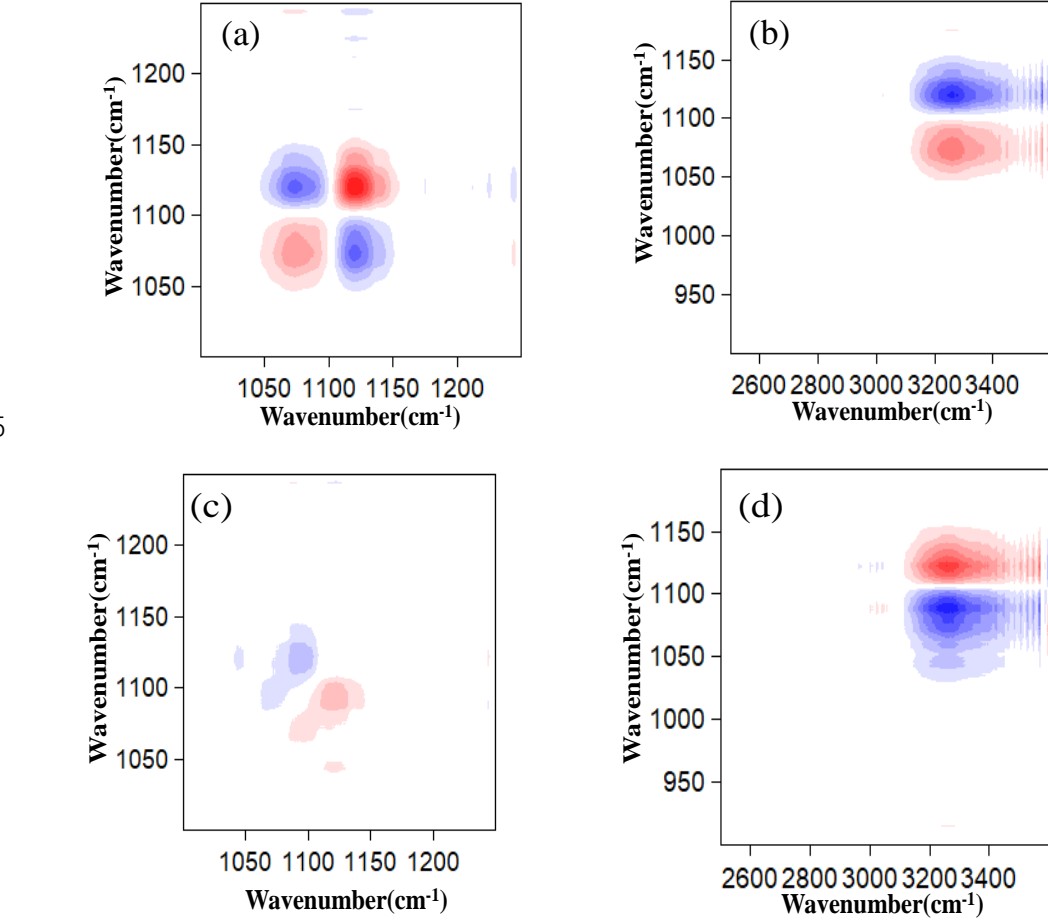

**Figure 7** (a) and (b) are synchronous correlation maps of AS nanoparticles within different wavenumber regions. (c) and (d) are the same as (a) and (b) but for asynchronous correlation maps. Red and blue areas represent positive and negative correlations, respectively.

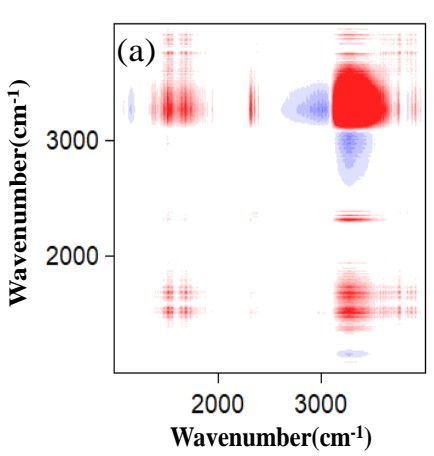
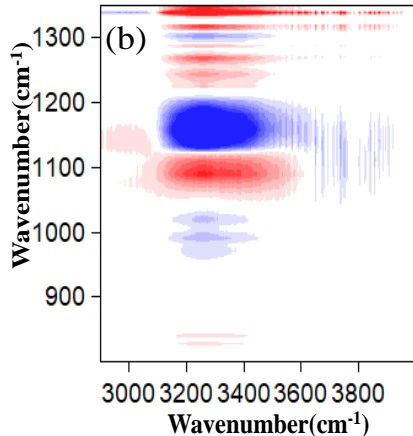

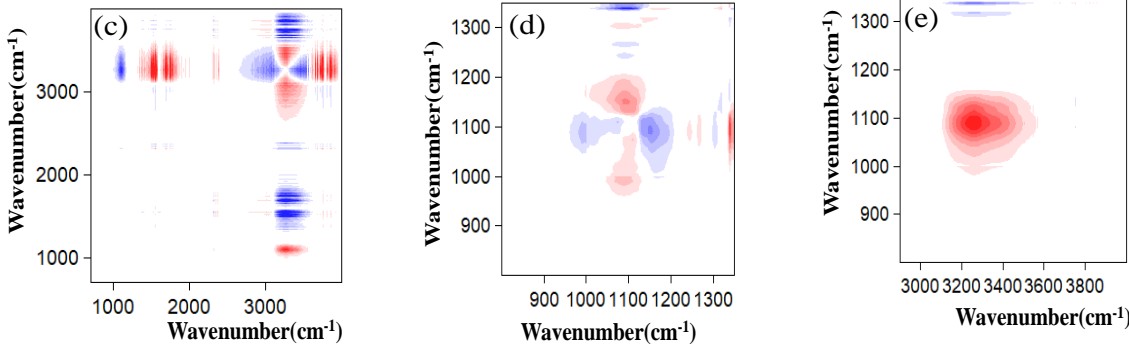

**Figure 8**. The same as Figure 7 but for AS/SN mixed nanoparticles. (b) is an enlarge view of (a) and (d) and (e) are enlarge views of (c) within different wavenumber regions.

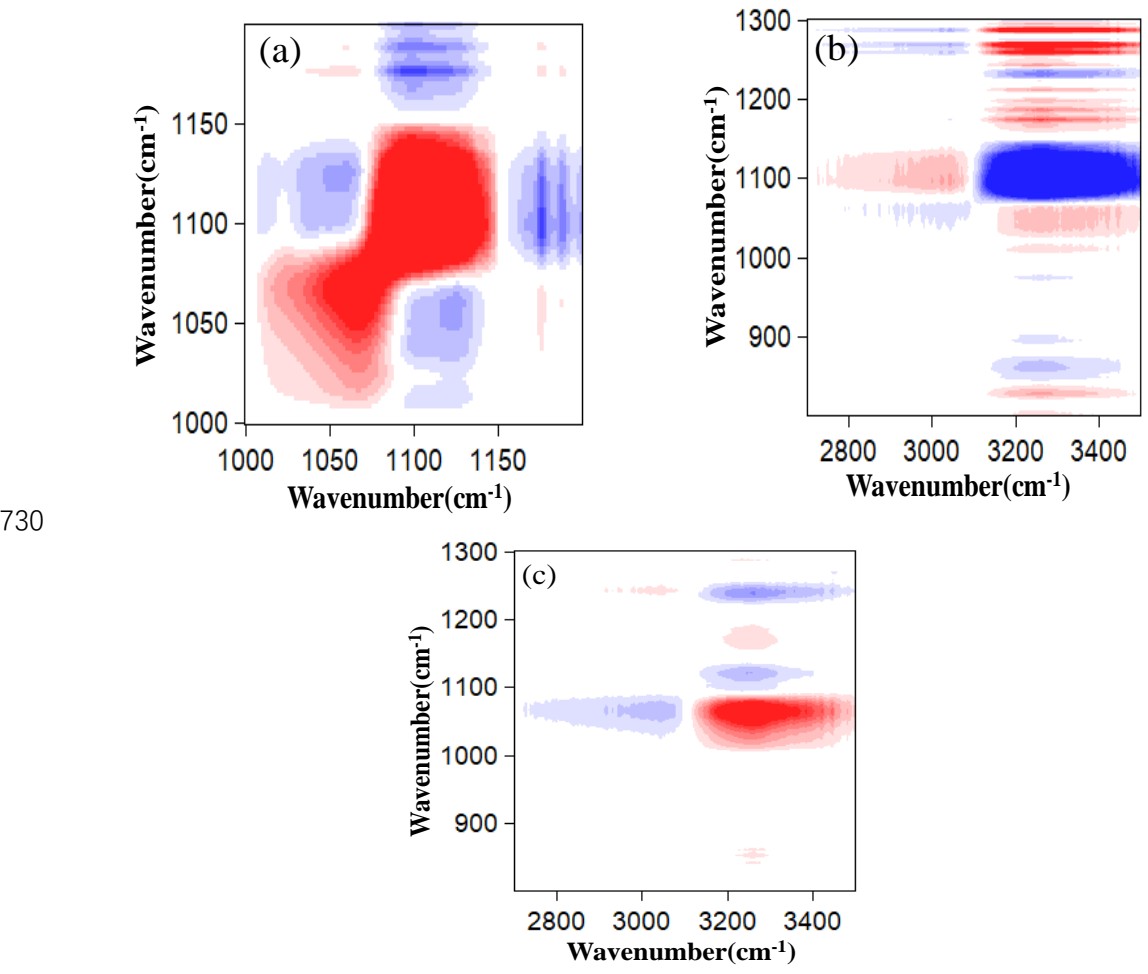

**Figure 9**. The same as Figure 7 but for AS/OA.