# Peer review of "Real-time diagnosis of the hygroscopic growth microdynamics of nanoparticles with Fourier transform infrared spectroscopy"

_Atmospheric Chemistry and Physics, 2021_

## Author Comment (AC1)

**Response to Referee #1:**

Thanks very much for your comments, suggestions and recommendation with respect to improve this paper. The response to all your comments are listed below.

In this manuscript, the authors present the hygroscopic study of 100 nm ammonium sulfate and ammonium sulfate mixed with sodium nitrate/oxalic acid nanoparticles using Fourier transform infrared (FTIR). The aerosol liquid water content was obtained using the FTIR spectral, and further the hygroscopic growth factor was calculated. The GF measurements are neither better nor worse than those of previous studies. The sequential order during the deliquescence process was also discussed with the 2D-IR spectroscopic analysis. From this point, the method is quite meaningful to better understand the intermolecular interaction within the phase transition.

The work is laboratory-based but has relevance to modeling. The topic of the manuscript fits into the scope of ACP. However, there are still some small mistakes in the manuscript. More discussion could be addressed before publication. I have several comments and suggestions for the authors below.

**Major comments:**

1. The authors mentioned the nanoparticle with approximately 100 nm is the electrical mobility diameter (line 25). Later the diameter of 100 nm was described as volume equivalent diameter in the experiment description section. The authors better describe clearly. For spherical particles, assuming that the particle and its mobility equivalent sphere have the same charge, then Dem=Dve. For non-spherical particles, the shape factor and slip correction factor should be considered.

**Response:** In our paper, we used a differential mobility analyzer (DMA) to sort the diameter of the nanoparticle. The diameter of 100 nm in the experiment description section should also be electrical mobility diameter. In the revised version, we have changed "volume equivalent diameter" to "electrical mobility diameter" in the experiment description in section 2.1. In section 2.3.2, we have proved that the Dem is close to Dve. Yan et al. (2020) have compared the  $D_{ve}$  and the  $D_{em}$  of AS sorted by the identical DMA of this study, which is shown in the figure R1. A good agreement between  $D_{em}$  and  $D_{ve}$  for ~100 nm AS was observed by Yan et al. (2020). As a result, in present work, we use  $D_{em}$  the same as the  $D_{ve}$ . For the AS with an electrical mobility

diameter of  $\sim 100$  nm, its volume equivalent diameter is  $\sim 94$  nm. Please check the marked up file for details.

**Figure R1 The SEM image of 100 nm AS particles deposited on the silicon wafer**

2. The authors stressed several times that the nanoparticle shape is one of the large uncertainties for the hygroscopicity study. Because normally the nanoparticle is assumed to have a spherical shape. Actually, the authors also supposed spherical particles when calculating the growth factor in this study, even the particles were deposited on the substrate. Since this is not a shape factor study, I would recommend do not emphasize this point.

**Response:** We have followed your suggestion and don't emphasize this point. We have changed some sentences in the abstract. Please check the marked up file for details.

**3. About RH:**

1). The authors claim the accuracy of RH measurement for the sample cell is 0.1% (Line 117). Is this 0.1% available for the whole studied RH range? Which sensor or instrument was employed? Normally the uncertainly will get larger when the RH increases (Mikhailov and Vlasenko, 2020). And 0.1% RH is extremely precise.

**Response**: This is a typing mistake. The sensor for RH measurement is HC2-S manufactured by Rotronic Incorporation (Switzerland), and its accuracy is  $\pm 0.8\%$  (RH range: 0  $\sim$  100%). In the revised version, we have corrected this mistake. Please check the marked up file for details.

2). The initial RH is 45% for AS measurement and there is an OH stretching vibration peak at 3250 cm-1 for all RH below DRH in Figure 2. Can authors explain why? I am considering whether the RH downstream of diffusion dryer (or DMA) was below the ERH value of AS. If not, then the particle is a droplet at the initial 45% RH.

**Response**: Since the FTIR instrument was not working under vacuum condition, OH stretching vibration peak at 3250 cm-1 for all RH below DRH in Figure 2 could be attributed to the liquid water in the ambient atmosphere and not the peak for NH4+ which locates at about 3000cm-1. Moreover, we found that the liquid water mass calculated from these peaks was constant, which further consolidate our deduction. In the revised version, we eliminated the interference of OH stretching vibration peak at 3250 cm-1 by subtracting the background and added more data points in Fig.2. As shown in Figure R2, the RH downstream of DMA was 16.52  $\sim$  18.74% which is below the ERH value (about ~32% RH) of AS.

Figure R2 Time series of RH downstream of DMA

4. The authors mentioned the mass of nanoparticles was quantified using a simple procedure. Could authors prove more details? And it would be nice to add some discussions about the uncertainty of mass and GF. Actually, there are error bars shown in some figures. But there is no description in the main text.

**Response:** We have included a new section, i.e., section 2.3.1, to present how we quantify the mass of nanoparticles and estimate the uncertainty. Briefly, the procedure is based on a non-linear least squares algorithm. It fits a measured spectrum by iteratively recalculating the spectrum until the mean-squared residual between the measured spectrum and calculated spectra are minimized. The calculated spectra are then the standard absorption spectra of liquid water. We also described the error bars in all figures. Please check the marked up file for details.

5. The authors illustrated the OA does not absorb water in the RH between 40% to 90%

in Figure 4. However, some previous studies demonstrated that oxalic acid dihydrate could form which will absorb water continually (Wang et al., 2017; Ma et al., 2019; Prenni et al., 2001). Since the FTIR could identify the hydration interactions, any explanations?

**Response:** Several previous measurements through the H-TDMA have found substantial growth of oxalic acid particles under low RH, which is mainly due to its amorphous state. In addition, it was reported that anhydrous oxalic acid particles could transform into oxalic acid dihydrate between 10 and 30% RH which can lead to a growth of size with the GF of around 1.17 during the transform. As a result, if the initial particles prepared can be in the form of oxalic acid dihydrate or nonstoichiometric hydrates containing about two water molecules per oxalic acid molecule, oxalic acid particles have no significant increase in the GF below 90% RH. Most stable dihydrate OA particles are easy to find in bulk studies but the OA particles in this study could be crystalline. Moreover, theoretical prediction and bulk measurements indicate that the deliquescence point of OA is greater than 97% RH (Peng et al., 2001).

In our paper, the initial OA particles prepared was in the form of oxalic acid dihydrate and there was no significant increase in the GF observed below 90% RH. If OA absorb water continually, the 3250cm-1 peak intensity will increase continually. We have included this interpretation in the revised version. Please check the marked up file for details.

6. The DRH point in both AS/NA and AS/OA systems is lower than the pure AS. According to 2D-IR results, the hydrolysis reaction mechanism of AS/OA is different from AS/NA. Also considering OA does not absorb water, then why the DRH value gets smaller?

**Response**: We have included the following interpretation in the revised version.

Although OA particle does not absorb water, OA and AS in the AS/OA aqueous solution can react with each other via the following pathway (Minambres et al., 2013):

 $(NH_4)_2SO_4 + H_2C_2O_4 \rightarrow NH_4HSO_4 + NH_4HC_2O_4$

This reaction can be identified in Figure S5, where the absorption peak at 1245cm-1 is the stretching vibration peak of HSO4-. As a result, a lower DRH for the mixed nanoparticles relative to the pure AS can be due to the formation of NH4HSO4 which has a lower DRH (about 40%) than the pure AS (80%) (Tang and Munkelwitz, 1994).

**Minor comments**

According to the detailed description of manuscript type in ACP (https://www.atmosphericchemistry- and-physics.net/about/manuscript\_types.html), I would recommend Research articles instead of Technical notes.

**Response:** We have included many description, comparison, analysis, and discussion in the revised version. As we have followed your recommendation and change it to a research article.

Line 38, "but only AN can change the hydrolysis reaction mechanism for AS in AS/AN and AS/OA mixtures." Is there AS/AN and AS/OA mixture experiment?

**Response:** Yes, in the experiments for AS/SN and AS/OA mixture, we found AN can change the occurrence sequential order for AS, but OA can not. We have added this explanation in the revised version. Please check the marked up file for details.

Line 42, "... between nanoparticle and medium", medium size particle? Please rewrite this sentence. Same Line 102

**Response:** Done. Please check the marked up file for details.

Line 47, "Nanoparticles have long atmospheric lifetimes of weeks to months". Any references?

**Response:** Done. We have included a reference for this description. Please check the marked up file for details.

Line 80, "the hygroscopic growth process of a single aerosol with particle diameter of less than 100 nm" maybe become "the hygroscopic growth process of a single particle with a diameter of less than 100 nm". Please rewrite this sentence.

**Response:** Done. Please check the marked up file for details.

Line 155, "room temperature is assumed to be 25 °C". What does this mean? No temperature sensor could measure the room temperature? Since the temperature has an influence on the RH, is there any temperature monitoring inside the sample cell?

**Response:** The air conditioner was run uninterruptedly to keep the laboratory under constant temperature. We put a temperature sensor in the room and the room temperature is a 25 °C. We thought the sample cell has the same temperature. There is no temperature monitoring inside the sample cell. We have included this description in the revised version. Please check the marked up file for details.

Line 156, the authors described the RH varies from 50% to 95% in the present work. But in AS study, the initial RH is 45%. In the OA study, the initial RH is 40%.

**Response:** Starting the initial RH from 40%, 45% or 50% does not affect the deliquescence point and deliquescence process because all of them are below the deliquescence point. To uniform the description, in all cases, we have changed the RH from 50% to 95% in the revised version. Please check the marked up file for details.

Line 209, "the nanoparticle volume increases but its mass keeps constant." Quite misleading, its means sulfate not nanoparticle, right?

**Response:** We mean that "As a further increase in RH, the volume of AS nanoparticles increases due to the increase in liquid water content, but the mass of AS nanoparticles keeps constant, resulting in a decrease in  $SO_4^{2-}$  concentration. As a result, we observe a decrease in the area of  $SO_4^{2-}$  absorption peak starting from ~ 83% RH.". We have changed this sentence in the revised version. Please check the marked up file for details.

Line 255, "at the RH of  $79.9 \pm 0.10\%$ ", is this RH measurement from this study? And how do the authors obtain this value? Since the FTIR measurement is a real-time method. Why authors don't provide more data points between 74% and 81% RH?

**Response:** We have included more data points between 74% and 81% RH (at a 2% RH interval) using FTIR measurement with different spectral resolution and repeat times. In this paper, more data could provide more information to understand the intermolecular interaction within the phase transition. "at the RH of  $79.9 \pm 0.10\%$ ", should be at the RH of  $79 \pm 0.80\%$ , which is the measurement from this study. We have revised accordingly in the revised version.

Line 282, what does the 1097 cm-1 stand for? NH4+? It seems hard to see a peak at 1097 cm-1 from Figure 2.

**Response:** Generally, 1097 cm-1 stands for sulfate in the aqueous phase. For  $NH_4^+$ , its peak shows in about 1442cm-1, and the strong peak of H2O (gas) show in 1350-1850 cm-1. So in the FTIR spectroscopy, we could not distinguish the  $NH_4^+$  peak.

Line 310, what does the 1320 cm-1 stand for? I would recommend the authors provide all FTIR spectral figures (OA, AS/NA, and AS/OA) at least in the supplement. Table1, I would recommend adding the reference (or data source) for density and solubility. **Response:** The 1320 cm-1 stands for the absorption of NO3- hidden by the H2O (gas) peaks in 1300-1900 cm-1. The NO3- peak in 1320cm-1 would change but the water (gas) peaks in 1300-1900 cm-1 keep roughly unchanged with the increase in RH. The 2D-IR analysis method can provide detailed information about the dynamic deliquescence processes of NO3- peak.

We have provided all FTIR spectral figures (OA, AS/NA, and AS/OA) in the supplement. And we have added the data source for density and solubility in Table1. Please check the marked up file for details.

Technical corrections

Unified abbreviation. Line 33 & 37, Sodium nitrate (SN); Line 38 & 90, sodium nitrate (AN); In the Figure 5&6, NN.

**Response:** Done. Please check the marked up file for details.

Line 52, "is" not are

**Response:** Done. Please check the marked up file for details.

Line 68, "the" nanoscale

**Response:** Done. Please check the marked up file for details.

Line 73, "the" nanoparticle

**Response:** Done. Please check the marked up file for details.

Line 92, "in" real time, not on real time; also other places

**Response:** Done. Please check the marked up file for details.

Line 93, "the" molecular scale

**Response:** Done. Please check the marked up file for details.

Line 209, "large" size particles, not big **Response:** Done. Please check the marked up file for details.

Line 212, "the" Kelvin effect

**Response:** Done. Please check the marked up file for details.

Line 238, "behavior" not behavior **Response:** Done. Please check the marked up file for details.

Line 247, "via direct measurement of the aerosol diameter" **Response:** Done. Please check the marked up file for details.

Line 259, "increasing"

**Response:** Done. Please check the marked up file for details.

Line 263, "the" FTIR measurement

**Response:** Done. Please check the marked up file for details.

Line 349 & 359, "the" 2D-IR spectroscopic

**Response:** Done. Please check the marked up file for details.

Line 362, "a" better understanding

**Response:** Done. Please check the marked up file for details.

Ma, Q., Zhong, C., Liu, C., Liu, J., Ma, J., Wu, L., and He, H.: A Comprehensive Study about the Hygroscopic Behavior of Mixtures of Oxalic Acid and Nitrate Salts: Implication for the Occurrence of Atmospheric Metal Oxalate Complex, ACS Earth and Space Chemistry, 3, 1216-1225, 10.1021/acsearthspacechem.9b00077, 2019.

Mikhailov, E. F., and Vlasenko, S. S.: High-humidity tandem differential mobility analyzer for accurate determination of aerosol hygroscopic growth, microstructure, and activity coefficients over a wide range of relative humidity, Atmos Meas Tech, 13, 2035-2056, 10.5194/amt-13-2035-2020, 2020.

Prenni, A. J., DeMott, P. J., Kreidenweis, S. M., Sherman, D. E., Russell, L. M., and Ming, Y.: The Effects of Low Molecular Weight Dicarboxylic Acids on Cloud Formation, J. Phys. Chem. A, 105, 11240-11248, 10.1021/jp012427d, 2001

---

## Author Response (AR1)

**Point-by-point response letter**

Note: We have included many descriptions, comparisons, analysis, and discussion in the revised version. Now we think this paper fits well within the scope of a research article (https://www.atmosphericchemistry-and-physics.net/about/manuscript_types.html). As a result, we have followed the recommendation of referee#1 and change this paper to a research article. We will be very grateful if the editor and the other two referees agree with this change. Otherwise, we are fine and will change back to technical note.

This file includes comments from three referees, the corresponding point-by-point responses, and the related changes in the manuscript. The black font are comments from the referees, and the red font are authors' responses as well as the related change clarifications.

**(1)Detailed response to comments from referee #1:**

In this manuscript, the authors present the hygroscopic study of 100 nm ammonium sulfate and ammonium sulfate mixed with sodium nitrate/oxalic acid nanoparticles using Fourier transform infrared (FTIR). The aerosol liquid water content was obtained using the FTIR spectral, and further the hygroscopic growth factor was calculated. The GF measurements are neither better nor worse than those of previous studies. The sequential order during the deliquescence process was also discussed with the 2D-IR spectroscopic analysis. From this point, the method is quite meaningful to better understand the intermolecular interaction within the phase transition.

The work is laboratory-based but has relevance to modeling. The topic of the manuscript fits into the scope of ACP. However, there are still some small mistakes in the manuscript. More discussion could be addressed before publication. I have several comments and suggestions for the authors below.

**Major comments:**

1. The authors mentioned the nanoparticle with approximately 100 nm is the electrical

mobility diameter (line 25). Later the diameter of 100 nm was described as volume equivalent diameter in the experiment description section. The authors better describe clearly. For spherical particles, assuming that the particle and its mobility equivalent sphere have the same charge, then Dem=Dve. For non-spherical particles, the shape factor and slip correction factor should be considered.

**Response:** In our paper, we used a differential mobility analyzer (DMA) to sort the diameter of the nanoparticle. The diameter of 100 nm in the experiment description section should also be electrical mobility diameter. In the revised version, we have changed "volume equivalent diameter" to "electrical mobility diameter" in the experiment description in section 2.1. In section 2.3.2, we have proved that the Dem is close to Dve. Yan et al. (2020) have compared the $D_{ve}$ and the $D_{em}$ of AS sorted by the identical DMA of this study, which is shown in the figure R1. A good agreement between $D_{em}$ and $D_{ve}$ for ~100 nm AS was observed by Yan et al. (2020). As a result, in present work, we use $D_{em}$ the same as the $D_{ve}$. For the AS with an electrical mobility diameter of ~100 nm, its volume equivalent diameter is ~ 94 nm. Please check page 4, line 110 and page 7, line 191-194 in the marked up file for details

[Figure]

Figure R1 The SEM image of 100 nm AS particles deposited on the silicon wafer

2. The authors stressed several times that the nanoparticle shape is one of the large uncertainties for the hygroscopicity study. Because normally the nanoparticle is assumed to have a spherical shape. Actually, the authors also supposed spherical particles when calculating the growth factor in this study, even the particles were deposited on the substrate. Since this is not a shape factor study, I would recommend do not emphasize this point.

**Response:** We have followed your suggestion and don't emphasize this point. We have changed some sentences in the abstract. Please check page 1, line 16-18 in the marked up file for details.

3. About RH:

1). The authors claim the accuracy of RH measurement for the sample cell is 0.1% (Line 117). Is this 0.1% available for the whole studied RH range? Which sensor or instrument was employed? Normally the uncertainly will get larger when the RH increases (Mikhailov and Vlasenko, 2020). And 0.1% RH is extremely precise.

**Response**: This is a typing mistake. The sensor for RH measurement is HC2-S manufactured by Rotronic Incorporation (Switzerland), and its accuracy is ±0.8% (RH range: 0 ～ 100%). In the revised version, we have corrected this mistake. Please check page 4, line 126-128 in the marked up file for details.

2). The initial RH is 45% for AS measurement and there is an OH stretching vibration peak at 3250 cm$^{-1}$ for all RH below DRH in Figure 2. Can authors explain why? I am considering whether the RH downstream of diffusion dryer (or DMA) was below the ERH value of AS. If not, then the particle is a droplet at the initial 45% RH.

**Response**: Since the FTIR instrument was not working under vacuum condition, OH stretching vibration peak at 3250 cm$^{-1}$ for all RH below DRH in previous Figure 2 could be attributed to the liquid water in the ambient atmosphere and not the peak for $NH_4^+$ which locates at about 3000cm$^{-1}$. Moreover, we found that the liquid water mass calculated from these peaks was constant, which further consolidate our deduction. In the revised version, we eliminated the interference of OH stretching vibration peak at 3250 cm$^{-1}$ by subtracting the background and added more data points in Fig.2. As shown in Figure R2, the RH downstream of DMA was 16.52 ～ 18.74% which is below the ERH value (about ~32% RH) of AS. Please check page 5, line 128-131 in the marked up file for details.

[Figure]

**Figure R2** Time series of RH downstream of DMA

4. The authors mentioned the mass of nanoparticles was quantified using a simple procedure. Could authors prove more details? And it would be nice to add some discussions about the uncertainty of mass and GF. Actually, there are error bars shown in some figures. But there is no description in the main text.

**Response:** We have included a new section, i.e., section 2.3.1, to present how we quantify the mass of nanoparticles and estimate the uncertainty. Briefly, the procedure is based on a non-linear least squares algorithm. It fits a measured spectrum by iteratively recalculating the spectrum until the mean-squared residual between the measured spectrum and calculated spectra are minimized. The calculated spectra are then the standard absorption spectra of liquid water. Please check page 6, line 160-176 in the marked up file for details.

We also described the error bars in all figures. Please check page 26, line 702-703 in the marked up file for details.

5. The authors illustrated the OA does not absorb water in the RH between 40% to 90% in Figure 4. However, some previous studies demonstrated that oxalic acid dihydrate could form which will absorb water continually (Wang et al., 2017; Ma et al., 2019; Prenni et al., 2001). Since the FTIR could identify the hydration interactions, any explanations?

**Response:** Several previous measurements through the H-TDMA have found substantial growth of oxalic acid particles under low RH, which is mainly due to its amorphous state. In addition, it was reported that anhydrous oxalic acid particles could transform into oxalic acid dihydrate between 10 and 30% RH which can lead to a growth of size with the GF of around 1.17 during the transform. As a result, if the initial particles prepared can be in the form of oxalic acid dihydrate or nonstoichiometric hydrates containing about two water molecules per oxalic acid molecule, oxalic acid particles have no significant increase in the GF below 90% RH. Most stable dihydrate OA particles are easy to find in bulk studies but the OA particles in this study could be crystalline. Moreover, theoretical prediction and bulk measurements indicate that the deliquescence point of OA is greater than 97%RH (Peng et al., 2001).

In our paper, the initial OA particles prepared was in the form of oxalic acid dihydrate and there was no significant increase in the GF observed below 90% RH. If OA absorb water continually, the 3250cm$^{-1}$ peak intensity will increase continually. We have included this interpretation in the revised version. Please check page 10, line 301-308 in the marked up file for details.

6. The DRH point in both AS/NA and AS/OA systems is lower than the pure AS. According to 2D-IR results, the hydrolysis reaction mechanism of AS/OA is different from AS/NA. Also considering OA does not absorb water, then why the DRH value gets smaller?

**Response**: We have included the following interpretation in the revised version. Although OA particle does not absorb water, OA and AS in the AS/OA aqueous solution can react with each other via the following pathway (Minambres et al., 2013):

$$(NH_4)_2SO_4 + H_2C_2O_4 \rightarrow NH_4HSO_4 + NH_4HC_2O_4$$

This reaction can be identified in Figure S5, where the absorption peak at 1245cm$^{-1}$ is the stretching vibration peak of $HSO_4^-$. As a result, a lower DRH for the mixed nanoparticles relative to the pure AS can be due to the formation of $NH_4HSO_4$ which has a lower DRH (about 40%) than the pure AS (80%) (Tang and Munkelwitz, 1994). Please check page 11, line 339-346 in the marked up file for details.

**Minor comments**

According to the detailed description of manuscript type in ACP (https://www.atmosphericchemistry- and-physics.net/about/manuscript_types.html), I would recommend Research articles instead of Technical notes.

**Response:** We have included many description, comparison, analysis, and discussion in the revised version. As a result, we have followed your recommendation and change it to a research article.

Line 38, "but only AN can change the hydrolysis reaction mechanism for AS in AS/AN and AS/OA mixtures." Is there AS/AN and AS/OA mixture experiment?

**Response:** Yes, in the experiments for AS/SN and AS/OA mixture, we found AN can

change the occurrence sequential order for AS, but OA can not. We have added this explanation in the revised version. Please check page14, line 414-420 and line 436-441 in the marked up file for details.

Line 42, "… between nanoparticle and medium", medium size particle? Please rewrite this sentence. Same Line 102

**Response:** Done. Please check page 2, line 39-44 in the marked up file for details.

Line 47, "Nanoparticles have long atmospheric lifetimes of weeks to months". Any references?

**Response:** Done. We have included a reference for this description. Please check page 2, line 48-49 in the marked up file for details.

Line 80, "the hygroscopic growth process of a single aerosol with particle diameter of less than 100 nm" maybe become "the hygroscopic growth process of a single particle with a diameter of less than 100 nm". Please rewrite this sentence.

**Response:** Done. Please check page 3, line 80-82 in the marked up file for details.

Line 155, "room temperature is assumed to be 25 °C". What does this mean? No temperature sensor could measure the room temperature? Since the temperature has an influence on the RH, is there any temperature monitoring inside the sample cell?

**Response:** The air conditioner was run uninterruptedly to keep the laboratory under constant temperature. We put a temperature sensor in the room and the room temperature is a 25 °C. We thought the sample cell has the same temperature. There is no temperature monitoring inside the sample cell. We have included this description in the revised version. Please check page 5, line 140-141 in the marked up file for details.

Line 156, the authors described the RH varies from 50% to 95% in the present work. But in AS study, the initial RH is 45%. In the OA study, the initial RH is 40%.

**Response:** Starting the initial RH from 40%, 45% or 50% does not affect the

deliquescence point and deliquescence process because all of them are below the deliquescence point. To uniform the description, in all cases, we have changed the RH from 50% to 90% in the revised version. Please check page 6, line 181-49, Figure2 and Figure 4 in the marked up file for details.

[Figure]

**Figure 2.** FTIR spectral characteristics of the AS nanoparticles under humidity conditions from 50% to 90% (RH).

[Figure]

**Figure 4.** The same as Figure 2 but for OA nanoparticles.

Line 209, "the nanoparticle volume increases but its mass keeps constant." Quite misleading, its means sulfate not nanoparticle, right?

**Response:** We mean that " As a further increase in RH, the volume of AS

nanoparticles increases due to the increase in liquid water content, but the mass of AS nanoparticles keeps constant, resulting in a decrease in $SO_4^{2-}$ concentration. As a result, we observe a decrease in the area of $SO_4^{2-}$ absorption peak starting from ~ 83% RH.". We have changed this sentence in the revised version. Please check page 10, Line 294-296 in the marked up file for details.

Line 255, "at the RH of 79.9 ± 0.10%", is this RH measurement from this study? And how do the authors obtain this value? Since the FTIR measurement is a real-time method. Why authors don't provide more data points between 74% and 81% RH?

**Response:** We have included more data points between 74% and 81% RH (at a 2% RH interval) using FTIR measurement with different spectral resolution and repeat times. In this paper, more data could provide more information to understand the intermolecular interaction within the phase transition. "at the RH of 79.9 ± 0.10%", should be at the RH of 79 ± 0.80%, which is the measurement from this study. We have revised accordingly in the revised version. Please check page 9, Line 281-282 and figure 3 in the marked up file for details.

[Figure]

**Figure 3.** (a) Predicted $M_{water}/M_0$ and the area of OH stretching peak as a function of RH; (b) The center wavenumber and the area of the symmetrical stretching vibration peak of $SO_4^{2-}$ in the aqueous AS nanoparticles as a function of RH. The black curves show the E-AIM predictions. The orange box indicates the deliquescence relative humidity point.

Line 282, what does the 1097 cm$^{-1}$ stand for? NH4+? It seems hard to see a peak at

1097 cm$^{-1}$ from Figure 2.

**Response:** Generally, 1097 cm$^{-1}$ stands for sulfate in the aqueous phase. For $NH_4^+$, its peak shows in about 1442cm$^{-1}$, and the strong peak of $H_2O$ (gas) show in 1350-1850 cm$^{-1}$. So in the FTIR spectroscopy, we could not distinguish the $NH_4^+$ peak. Please check page 9, Line 266-267 and figure 3 in the marked up file for details.

Line 310, what does the 1320 cm$^{-1}$ stand for? I would recommend the authors provide all FTIR spectral figures (OA, AS/NA, and AS/OA) at least in the supplement. Table1, I would recommend adding the reference (or data source) for density and solubility.

**Response:** The 1320 cm$^{-1}$ stands for the absorption of $NO_3^-$ hidden by the $H_2O$ (gas) peaks in 1300-1900 cm$^{-1}$. The $NO_3^-$ peak in 1320cm$^{-1}$ would change but the water (gas) peaks in 1300-1900 cm$^{-1}$ keep roughly unchanged with the increase in RH. The 2D-IR analysis method can provide detailed information about the dynamic deliquescence processes of $NO_3^-$ peak. Please check page 13, line 400-401 in the marked up file for details.

We have provided all FTIR spectral figures (OA, AS/NA, and AS/OA) in the supplement. And we have added the data source for density and solubility in Table1. Please check page 10, line 299-301 and Figure S2, Figure S3, Figure S4 in the marked up file for details.

We have added the reference for density and solubility. Please check page 5, line 146-148 in the marked up file for details.

[Figure]

**Figure S2.** FTIR spectral characteristics of the SN nanoparticles under humidity

conditions from 55% to 85% (RH).

[Figure]

**Figure S3.** FTIR spectral characteristics of the AS/SN nanoparticles under humidity conditions from 50% to 90% (RH).

[Figure]

**Figure S4.** FTIR spectral characteristics of the AS/OA nanoparticles under humidity conditions from 55% to 88% (RH).

Technical corrections

Unified abbreviation. Line 33 & 37, Sodium nitrate (SN); Line 38 & 90, sodium nitrate (AN); In the Figure 5&6, NN.

**Response:** Done. Please check Page 2, Line 35, and Figure 5&6 in the marked up file for details.

[Figure]

**Figure 5.** Comparison between the measured and predicted $M_{water}/M_0$ for pure AS (a), AS/OA (b), and AS/SN nanoparticles (c) with a dry diameter of 100 nm during the hygroscopic growth process as a function of the RH. The black curves represent the E-AIM predictions and the red square represent the measured results from the FTIR spectra. The error bar is defined as the 1 σ standard deviation of repeated measurements.

[Figure]

[Figure]

**Figure 6** The same as Figure 5 but for GFs.

Line 52, "is" not are

**Response:** Done. Please check page 2, line 54 in the marked up file for details.

Line 68, "the" nanoscale

**Response:** Done. Please check page 3, line 69 in the marked up file for details.

Line 73, "the" nanoparticle

**Response:** Done. Please check page 3, line 74 in the marked up file for details.

Line 92, "in" real time, not on real time; also other places

**Response:** Done. Please check page 4, line 94 in the marked up file for details.

Line 93, "the" molecular scale

**Response:** Done. Please check heck page 4, line 93 in the marked up file for details.

Line 209, "large" size particles, not big

**Response:** Done. Please check page 9, line 281 in the marked up file for details.

Line 212, "the" Kelvin effect

**Response:** Done. Please check page 9, line 285 in the marked up file for details.

Line 238, "behavior" not behavior

**Response:** Done. Please check page 12, line 365 in the marked up file for details.

Line 247, "via direct measurement of the aerosol diameter"

**Response:** Done. Please check page 11, line 325 in the marked up file for details.

Line 259, "increasing"

**Response:** Done. Please check the marked up file for details.

Line 263, "the" FTIR measurement

**Response:** Done. Please check page 11, line 327 in the marked up file for details.

Line 349 & 359, "the" 2D-IR spectroscopic

**Response:** Done. Please check page 15, line 453 in the marked up file for details.

Line 362, "a" better understanding

**Response:** Done. Please check page 15, line 478 in the marked up file for details.

**(2)Detailed response to comments from referee #2:**

In this study, the FTIR spectroscopy and two-D correlation analysis were used to investigate the hygroscopic behavior of typical aerosols. It demonstrated that the method is good at qualifying and quantifying the interaction between water and aerosol. In particularly, the 2-D correlation analysis can provide more detail information about the hydration process. Thus, this method is helpful to understand the hygroscopicity of aerosols and it is suitable for publication as a technical note. However, the comments of referee #2 are also my concerns, and I have some other concerns as following.

**Response:** All your comments listed below have been addressed. Please check the point by point response as follows.

1. Since the samples were deposited on a ZnSe plate in stacking state, can they also be considered as nanoparticles? Further experiments using samples prepared by deposited solution with further dry process were recommended. Then the results of these two different preparation methods would be compared.

**Response:** The chemical composition of the 100nm nanoparticles is not changed during depositing onto the stacking state. And for the 100nm nanoparticles, its hydration characteristic mainly depends on its chemical composition and the kelvin effect is negligible. The enrichment for the nanoparticles is to improve the signal of FTIR because the hygroscopic signal of a single particle is too weak to be measured by the FTIR method. As a result, we believe this experiment is enough and no further experiments are needed. We have included this explanation in the revised paper. Please check page 4, line 119-125 in the marked up file for details.

2. Line 305, what's "surface-limited process"? Since all the initial step of particle hydration could be water adsorption on surface, surface limited process should always determine the hygroscopic behavior.

**Response:** Surface-limited process means the surface limited process could determine the hygroscopic behavior. The rate of hydrolysis depends on the water content from the surface into the substance. In our study, the decrease of $SO_4^{2-}$ in the solid AS nanoparticles does not result in a simultaneous increase of $SO_4^{2-}$ in the aqueous AS nanoparticles during the AS deliquescence transition. The former process occurred predominantly before the later process. It suggests an intriguing possibility of the existence of an intermediate state between the solid and aqueous AS nanoparticles. Meanwhile, the water uptake process occurred predominantly before the decrease of $SO_4^{2-}$ in the solid AS nanoparticles during the AS deliquescence transition. We speculate that the surface-limited process may control the transport of liquid water to the AS nanoparticle, or in other words, the surface limited process determine the hygroscopic behavior of AS nanoparticles (Leng et al., 2015).

We have added this explanation in the revised version, please check page 12, line 378-382 the marked up file for details.

3. In the previous study of wang et al. (2017), the formation of ammonium hydrogen oxalate ($NH_4HC_2O_4$) and ammonium hydrogen sulfate ($NH_4HSO_4$) from interactions between OA and AS in aerosols during the dehydration process were observed. Did you detect these species in the present study? Could the 2-D correlation analysis confirm this reaction?

**Response:** In our experiment, the peak of $HC_2H_4^-$ shows in the $1245 cm^{-1}$. Because the particles are aerosolized by an atomizer (model 255, MetOne) and dried by a diffusion dryer (model 3062, TSI). The RH would decrease to 25% under this dehydration process. But the mixture OA/AS in our experiment was in the hydration process, so we did not measure the process for the formation of ammonium hydrogen oxalate ($NH_4HC_2O_4$) and ammonium hydrogen sulfate ($NH_4HSO_4$) from interactions between OA and AS in aerosols. But during the dehydration process, the 2-D correlation analysis confirm this reaction due to the peak for the $HSO_4^-$ or $HC_2O_4^-$. Please check page 11, line 339-346 the marked up file for details.

(3)**Detailed response to comments from referee #3:**

This study used the FTIR spectroscopy and two-D correlation analysis to investigate the hygroscopicity of several single-component aerosols and their mixtures. The sequential order during the hydration process was discussed with the 2D-IR spectroscopic analysis. The method is new. However, as a technical note, validation of the performance of current method should be conducted in a more thorough way. I have several comments regarding current version of the manuscript and hope the authors could carefully address those before being considered for publication.

**Response:** All your comments listed below have been addressed. Please check the point by point response as follows.

**Major comments**

Since submitted as a technical note, the authors should provide a solid and sound validation of the accuracy and stability of their technique. For instance, was your DMA calibrated?

**Response:** Yes, we have calibrated. As shown in Figure R1, the SEM-measured particle size is consistent with the DMA screening size (about 100 nm). Please check page 7, line 191-194 the marked up file for details.

[Figure]

Figure R1 The SEM image of 100 nm AS particles deposited on the silicon wafer

How much did you dry your aerosols?

**Response:** As shown in Figure R2, the RH downstream of DMA was 16.52 ～ 18.74% which was below the ERH value (about ~32% RH) of AS. We have included this sentences in the revised version: The RH downstream of the DMA varies over 16.52 to 18.74% which are well below the efflorescence relative humidity (ERH) of AS (about ~32% RH) (Figure R2). As a result, the initial states of all nanoparticles are in dry conditions. Please check page 5, line 129-131 the marked up file for details.

[Figure]

**Figure** R2 Time series of RH downstream of DMA

How many particles did you deposit onto the substrate before further measurements, would the concentration of the particles deposited influence your results?

**Response:** There were about 100 thousand particles deposited onto the substrate. The measured particles deposited onto the substrate could overlap on each other for FTIR. But the chemical composition is not changed for the 100nm nanoparticles during depositing onto the stacking state. And for the 100nm nanoparticles, its hydration characteristic mainly depends on its chemical composition and the kelvin effect is negligible. The enrichment for the nanoparticles is to improve the signal of FTIR because the hygroscopic signal of a single particle is too weak to be measured by the FTIR method. Since the chemical composition is not changed for the 100nm nanoparticles during depositing onto the stacking state, the concentration of the particles deposited does not influence our results. We have included this explanation in the revised paper. Please check page 4, line 116-125 the marked up file for details.

How was the RH measured, what was its uncertainty?

**Response:** The RH was measured with a RH sensor HC2-S manufactured by Rotronic Incorporation (Switzerland), and its accuracy was ±0.8% (RH range: 0 ～ 100%). We have included this description in the revised version. Please check page 5, line 126-129 the marked up file for details.

Was the cell well-insulated that the RH could be maintained?

**Response:** Yes, the cell is well-insulated to maintain the RH at a constant level. A RH sensor was mounted at the inlet of the cell and the cell length is only ~ 4cm. The FTIR spectrometer starts to take absorption spectra of the samples approximately 5 min after the injection of each designated RH. This time interval is used to stabilize the RH inside the sample cell. We also measure the RH at the outlet of the cell and found that it is close to that at the inlet. Please check page 5, line 132-135 the marked up file for details.

What parameters did you directly obtain from the technique and how did you interpret them? Each single part of your instrument should be described clearly.

**Response:** Done. We can directly obtain the absorbance spectroscopy under different RH condition. ① We first correct the baseline of the measured absorption spectra with the Opus 7.0 software. We then iteratively recalculate the spectra with the absorption coefficients of liquid water provided by Downing and Williams (Downing and Williams, 1975) through the non-linear least squares method till the residual between the measured spectrum and the calculated spectra are minimized. We stop the iteration and derive the liquid water content once the root mean square error (RMSE) of the residual is below 0.3%. After the deliquescence of the nanoparticles and the position of the absorption peak (referring to the wavenumber that shows the strongest absorption) of $SO_4^{2-}$ is relative stable, we use a similar non-linear least square method to derive the mass of the sulfate ($M_{sulfate}$) (Wei et al., 2019). As a result, the mass of AS ($M_{AS}$), AS/SN ($M_{AS/SN}$), and AS/OA ($M_{AS/OA}$) can be derived with the $M_{sulfate}$ via the following equations.

$$M_{AS} = \frac{M_{sulfate}}{96} * 132 \tag{1}$$

$$M_{AS/SN} = \frac{M_{sulfate}}{96} * 132 * 2 \tag{2}$$

$$M_{AS/OA} = (\frac{M_{HSO_4^-}}{97} + \frac{M_{sulfate}}{96}) * 132 * 2 \tag{3}$$

The mass is used to calculate the hygroscopic growth factor (GF). Please check page 6, line 161-176 the marked up file for details.

②After baseline correction, the absorbance spectra are normalized into 2D-IR spectra with the 2D Shige software which can give the Synchronous and Asynchronous correlation maps. The red and blue areas in the 2D-IR spectra indicate positive and negative correlations of the spectral intensities at $v_1$ and $v_2$, respectively. In the synchronous correlation maps, the positive and negative correlations indicate simultaneous and opposite changes of the spectral intensities observed at the wavenumber pair ($v_1$, $v_2$), respectively. In the asynchronous correlation maps, the positive correlation indicates the spectral intensity change at $v_1$ occurs predominantly before that at $v_2$, while the negative correlation indicates the spectral intensity change at $v_2$ occurs predominantly before that at $v_1$.

We have added above descriptions in section 2.3.3. Please check page 7, line 204-243 in the marked up file for details.

Second, the authors used AS to calibrate the instrument. However, the figures the authors plotted are lack of critical data points, particularly between 74% RH and 81% RH.

**Response:** Done. We have included more data points in Fig.2 and Fig. 3. Please check the marked up file for details.

[Figure]

**Figure 2.** FTIR spectral characteristics of the AS nanoparticles under humidity conditions from 50% to 90% (RH).

[Figure]

**Figure 3.** (a) Predicted $M_{water}/M_0$ and the area of OH stretching peak as a function of RH; (b) The center wavenumber and the area of the symmetrical stretching vibration peak of $SO_4^{2-}$ in the aqueous AS nanoparticles as a function of RH. The black curves show the E-AIM predictions. The orange box indicates the deliquescence relative humidity point.

The authors deposited 100 nm particles onto a substrate. These material could not represent nanoparticles anymore. The condensation of water molecules on a curved particle could be much different from that on a flat surface with deposited substances on. The author should clarify these differences and give proper discussion, emphasizing its own atmospheric relevance. A section discussing its atmospheric implications is suggested.

**Response:** The measured particles deposited onto the substrate could overlap on each other for FTIR. But the chemical composition is not changed for the 100nm nanoparticles during depositing onto the stacking state. And for the 100nm nanoparticles, its hydration characteristic mainly depends on its chemical composition and the kelvin effect is negligible. The enrichment for the nanoparticles is to improve the signal of FTIR because the hygroscopic signal of a single particle is too weak to be measured by the FTIR method. Since the chemical composition is not changed for the 100nm nanoparticles during depositing onto the stacking state, the concentration of the particles deposited does not influence our results. Lee et al., concluded that, for particles with volume equivalent diameter of greater than 100nm, their critical hydration characteristics are essentially independent of the particle size, which is similar to the condensation of water on the flat surface (Lee et al., 1998) . In the

revised version, we have given the explanation, Please check page 10, line 284-289 in the marked up file for details.

The authors stated in their abstract as well as in the intro part that: "current techniques are also difficult to identify the intermolecular chemical interactions of phase transition micro-dynamics during nanoparticle deliquescence process because their limited temporal resolutions are unable to capture the complex femtosecond-level intermediate states". I am curious from current results you presented did not show a higher temporal resolutions and femtosecond-level intermediate states were not displayed, either. Otherwise, please clarify.

**Response:** Done. In our manuscript, the femtosecond-level temporal resolutions are not displayed, but the intermediate states could be indicated from the he IR peaks change in the order. We have revised this sentence in the version. Please check page 1, line 19-23 in the marked up file for details.

Based on my third comment, the author should emphasis the novelty of current technique. What are the advantages compared to traditional hygroscopicity measurements? I assume Sect 3.3 and 3.4 are quite important, contributing new insights to current understandings. However, the discussion in these parts was kind of weak and plain. Proper comparisons with other studies may help.

**Response**: Compared to traditional hygroscopicity measurements, the 2D-IR spectroscopic technique captured more complex process and the intermediate state during the hygroscopic growth of AS nanoparticles (Tang, et al., 1977; Tang, et al. 1994). In the revised version, we have presented proper comparisons and analysis with previous studies in section 3.3 and 3.4. Although the hygroscopic growth characteristics observed in this study are similar to those in previous study (Cruz and Pandis, 2000), the 2D-IR spectroscopic technique captured more complex process and the intermediate state during the hygroscopic growth of AS nanoparticles. Please check page 13, line 383-393 and page 14, line 415-421, line 437-442 in the revised version for details.

**Minor comments:**

Line 90: AN or SN? Similar in line 96.

**Response:** Done. Please check page 4, line 97 in the marked up file for details.

Line 102: Current work is of great significance, not only for the haze control across China, but also for the whole atmospheric community. There remains huge gaps and requires significant efforts for the haze control across China using current results. Consider rephrase the sentence or extend the content.

**Response:** We have tempered this description in the revised version. Please check page 4, line 103-104 in the marked up file for details.

Line 108: How much you dried your aerosols? For instance, to which RH condition?

**Response:** We have measured the RH downstream of DMA and found they varied over 16.52 to 18.74%, which are all below the ERH value (about ~32% RH) of AS . We have included this description in the revision: The RH downstream of the DMA varies over 16.52 to 18.74% which are well below the efflorescence relative humidity (ERH) of AS (about ~32% RH) (Figure R2). As a result, the initial states of all nanoparticles are in dry conditions. Please check page 5, line 129-131 the marked up file for details.

[Figure]

**Figure R3** Time series of RH downstream of DMA

Line 110: How many particles you deposited onto the substrate? Will they overlap on each other?

**Response:** There were about 100 thousand particles deposited onto the substrate, and they could overlap on each other. But the chemical composition is not changed for the 100nm nanoparticles in the stacking state. And for the 100nm nanoparticles, its hydration character depends on mainly its chemical composition, and the kelvin effect is negligible. The enrichment for the nanoparticles is to improve the signal of FTIR because the hygroscopic signal of a single particle is too weak to be measured by the FTIR method. We have included this description in the revision. Please check page 4, line 116-126 in the marked up file for detail.

Line 151-160: Even though you described our instrumentation, I am still confused how you measure or derive Dwet.

**Response:** The Dwet was not measure directly. It was calculated through the increased water mass and $D_0$. $D_0$ (cm) is the mean initial diameter of the dry particles which was 100nm. When the RH varies from 50% to 95%. The total water mass could be calculated in different RH conditions. For example, for pure AS, we can calculate the initial mass of the dry AS particles through the simple procedure base on the sulfate IR absorbance. The simple procedure is non-linear least squares algorithm. It means that a measured spectrum is fitted by iteratively recalculating the spectrum until the mean-squared residual between the measured spectrum and calculated spectra is minimized. The calculated spectra are the standard absorption spectrum of liquid water and AS. So from the $V_0$ and $V_{wet}$ can be calculated via equations (1) and (2). So the Dwet could be derived vis equations (1) to (4).

$$V_{water} = \frac{M_{water}}{\rho_{water}} \tag{1}$$

$$V_0 = \frac{M_{AS}}{\rho_{AS}} \tag{2}$$

$$V_{wet} = V_0 + V_{water} \tag{3}$$

$$D_{wet} = (\frac{3V_{wet}}{4\pi})^{1/3} \tag{4}$$

Where $V_0$ (cm$^3$) is the initial volume of the dry nanoparticle at approximately 25°C, and $V_{water}$ (cm$^3$) is the water volume contained in the nanoparticle at the designated

RH; $M_{water}$ (g) and $M_{AS}$ (g) are the calculated water mass and the pure AS component mass at the designated RH, respectively; $\rho_{water}$ (g/cm$^3$) (approximately 1 g/cm$^3$) and $\rho_{AS}$ (g/cm$^3$) are the densities of water and the pure AS component, respectively. We have included above detailed descriptions in the revised version. Please check page 6, line 161-196 the marked up file for details.

Line 193: What do you mean by "area of OH"?

**Response:** "area of OH" mean the area of the OH absorption peak at 3250 cm$^{-1}$. We have included this interpretation in the revised version. Please check page 8, line 249-252 in the marked up file for details.

Line 211: Your DRH value for small particles is similar to that of large ones from other studies. But this is not enough to prove the accuracy of your method, as GF value of AS at a certain RH is also a necessity and it varies between small particles and large ones. Rephrase the sentence and make thorough comparison.

**Response:** Our method and the size of particle are different from previous studies, but we obtained a consistent DRH (about 79%±0.8%) to those in Tang et al. (1982), Cruz and Pandis (2000), and Estillore et al. (2016) who measured a DHR for AS was 79%±1% and 80%±0.4%. Though the particle size (~100 nm) in this study is smaller than those in previous studies, its hydration characteristic depends on mainly its chemical composition and the kelvin effect is negligible. Moreover, Lee et al. (1998) concluded that, for particles with diameter greater than 100nm, their critical hydration characteristics are essentially independent of the particle size, which is similar to the condensation of water on the flat surface. We have included above description in the revision. Please check page 9, line 284-289 in the marked up file for details.

Line 211, also for Fig.2 and Fig. 3: Moreover, I observed that you performed the measurements at 74% RH and at 81% RH. At 81% RH, you observed deliquesence and you reported this value as DRH. This is not sound, it could be any values between 74% and 81% RH, please consider add more data points between these two RHs, specifically for your Fig.2 and Fig. 3. Before you add more data points at those RHs, I cannot be convinced.

**Response:** Done. We have included more data points in Fig.2 and Fig. 3. Please check

[Figure]

**Figure 2.** FTIR spectral characteristics of the AS nanoparticles under humidity conditions from 50% to 90% (RH).

[Figure]

**Figure 3.** (a) Predicted $M_{water}/M_0$ and the area of OH stretching peak as a function of RH; (b) The center wavenumber and the area of the symmetrical stretching vibration peak of $SO_4^{2-}$ in the aqueous AS nanoparticles as a function of RH. The black curves show the E-AIM predictions. The orange box indicates the deliquescence relative humidity point.

Line 218: What do you mean by 'nanoparticle volume increases but its mass keeps constant'? Please clarify.

**Response:** We mean that the nanoparticle volume increases as the increase in liquid water mass, but the AS mass keeps constant. We have changed this sentence to " As a further increase in RH, the volume of AS nanoparticles increases due to the increase in liquid water content, but the mass of AS nanoparticles keeps constant, resulting in a decrease in $SO_4^{2-}$ concentration. As a result, we observe a decrease in the area of $SO_4^{2-}$ absorption peak starting from ~ 83% RH." in the revised version. Please check page 10, line 294-296 in the marked up file for details.

Line 223: When you referred the calculation to previous section, for instance, $M_{water}$ and M0, you should specify all of these quantities in previous part. I am quite confused that how you obtained Mwater, Mwet-Mo?

**Response:** In the revised version, we have included detailed quantitative analysis for Mwater, $M_0$ in section 2.3.1. we also analyze the RMSE for the quantitative method. Please check page 5, line 161-176 in the marked up file for details.

Line 228: Please add comparison with other studies, especially 100 nm particles of oxalic acid. Boreddy et al. (2018) measured the GF of oxalic acid using a HTDMA and they found that oxalic acid started to absorb water at RH < 45% and the GF of oxalic acid was 1.47 at 90% RH. Please consider giving proper discussion.

**Response:** Several previous measurements using the H-TDMA found substantial growth of oxalic acid particles under low RH, which was mainly due to its amorphous state. In addition, it was reported that anhydrous oxalic acid particles could transform into oxalic acid dihydrate between 10 and 30% RH which could lead to a growth of size with the GF of around 1.17 during the transform. Thus, if the initial particles prepared could be in the form of oxalic acid dihydrate or nonstoichiometric hydrates containing about two water molecules per oxalic acid molecule, oxalic acid particles have no significant increase in the GF below 90% RH. Most stable dihydrate of OA particles are easy to find in bulk studies. Moreover, theoretical prediction and bulk measurements indicate the deliquescence point of OA is greater than 97%RH (Peng et al., 2001).

In our paper, the initial OA particles prepared was in the form of oxalic acid dihydrate and there was no significant increase in the GF observed below 90% RH. If OA absorb water continually, the 3250cm$^{-1}$ peak intensity will increase continually. In the revised version, we have given the detailed description. Please check page 10, line 200-308 in the marked up file for details.

Line 236: I suggest you add the results for sodium nitrate (SN) particles before you discuss the results for mixtures. Again, it should be SN but not AN, isn't it? Also in Fig. 5, what does NN stand for?

**Response:** Done. We have changed the abbreviation of sodium nitrate to SN and increase the FTIR spectral characteristics of the SN nanoparticles under humidity conditions from 55% to 85% (RH) in Supplement. Please check page 10, line 200-301 and Figure S2 in the marked up file for details.

[Figure]

**Figure S2.** FTIR spectral characteristics of the SN nanoparticles under humidity conditions from 55% to 85% (RH).

Line 254-259: Similarly, add more data points for GF at RH between 74% and 81% to complete your validation. And the content of this paragraph was well known by hygroscopicity community, you don't have to repeat it, but give detailed description of your DRH value as well as corresponding GF and compare with other studies.

**Response:** We have included the more data points in Fig.2 and Fig. 3. Furthermore, we have given detailed description of our DRH value as well as corresponding GF and compared with other studies. Please check page 11, line 331-336 in the marked up file for details.

Line 260-265, also for Fig. 6: At RH between 70% to 85%, your results for mixtures deviate from the predictions. Give proper explanations. You cannot just state that they are in good agreement with other studies without presenting their data points.
**Response:** Done. We have included more explanation in the revised version. Please check page 11, line 339-346 in the marked up file for details.

Line 269: What does absorption peak mean? Please clarify.
**Response:** Absorption peak refers to the wavenumber that shows the strongest absorption by a chemical sample which is used to identify specific radicals or compounds. Please check page 6, line 169-170 in the marked up file for details.

Line 271: Since 2D-IR spectroscopic technique is a major instrument for your results, you should describe it in a more detailed way. For instance, in Sect2.1. The description was too simplified and should be extended. Give possible examples and emphasis its significance. I do think line 267-277 should go to the experiment section. And please further explain "Synchronous correlation maps" and "asynchronous correlation maps", give examples and how we use these results, what do these results indicate or how we interpret them.
**Response:** Done. We have interpret them in section 2.3.3. Please check page 8, line 204-243 in the marked up file for details.

Line 310: So all the auto-peaks were all the same during all the RH conditions? Similar questions to the asynchronous correlation maps.
**Response:** The 2D-IR spectra represent the perturbation-induced variations of a series of spectral intensity observed during the interval of external variable RH. In present work, the red and blue areas in the 2D-IR spectra indicate positive and negative correlations of the spectral intensities at $v_1$ and $v_2$, respectively. In the synchronous

correlation maps, the positive and negative correlations indicate simultaneous and opposite changes of the spectral intensities observed at the wavenumber pair ($v_1$, $v_2$), respectively. In the asynchronous correlation maps, the positive correlation indicates the spectral intensity change at $v_1$ occurs predominantly before that at $v_2$, while the negative correlation indicates the spectral intensity change at $v_2$ occurs predominantly before that at $v_1$.

In the revised version, we have given the detailed description in section2.3.3, section 3.3 and section 3.4. Please check page 8, line 204-243 in the marked up file for details.

Line 341-345: Too long sentence, please rephrase it.

**Response:** Done. Please check page 14, line 444-454 the marked up file for details.

**Reference:**

Yuh-Lang Lee, Wen-Sheng Chou, Liang-Huei Chen, surface science, 1998, 414, 363-373